J Physiol 603.19 (2025) pp 5613–5628

5613

# Contusive spinal cord injury causes Nav1.8 dysfunction to upregulate small sensory neuron excitability

Yucheng Xiao[1], Yanling Pan[1] (iD), Naikui Liu[2] (iD) and Theodore R. Cummins[1,3] (iD)

[1]Department of Biology, School of Science, Indiana University-Indianapolis, Indianapolis, IN, USA
[2]Spinal Cord and Brain Injury Research Group, Stark Neurosciences Research Institute, Department of Neurological Surgery, Indiana University School of Medicine, Indianapolis, IN, USA
[3]Stark Neurosciences Research Institute, Indiana University School of Medicine, Indianapolis, IN, USA

Handling Editors: Nathan Schoppa & Carole Torsney

The peer review history is available in the Supporting Information section of this article (https://doi.org/10.1113/JP288215#support-information-section).

The Journal of Physiology

**Abstract figure legend** Contusive SCI significantly dysregulated excitability of small DRG neurons by enhancing Nav1.8 function. SCI-induced hyperexcitability was partially reversed by ZL0177, a small compound disrupting FHF interaction with Nav1.8 and Nav1.9. SCI: spinal cord injury; DRG: dorsal root ganglion; Nav: voltage-gated sodium channels; FHFs: fibroblast growth factor homologous factors. Created with Biorender.

**Abstract** Chronic neuropathic pain associated with spinal cord injury (SCI) poses a significant medical challenge. Studies have shown that moderate thoracic (T10) SCI can substantially enhance the excitability of small diameter dorsal root ganglion (DRG) neurons, suggesting that pain resulting from SCI may arise from pathological alternations in peripheral neurons. However, the molecular mechanisms underlying these changes remain unclear. Here we show that contusive SCI significantly

**Yucheng Xiao** is an Associate Research Professor in the Department of Biology at School of Science, Indiana University, Indianapolis. He is interested in understanding the molecular mechanisms that control neuronal excitability and disclosing the roles of ion channels in neurological disorders. He is also interested in discovering novel modulators of ion channels that regulate pain sensations and cardiovascular functions. **Yanling Pan** holds a PhD in Medical Neuroscience. Her current research focuses on applying computational and bioinformatics tools to study the progression of Alzheimer's disease.

Y. Xiao and Y. Pan contributed equally to this work.

increases transient and resurgent sodium currents mainly through Nav1.8 in rat small DRG neurons. Additionally, SCI increases the proportion of small DRG neurons that produce tetrodotoxin-resistant resurgent currents. The SCI-induced increase of Nav1.8 currents can be reversed by ZL0177, a small peptidomimetic of the critical residues in fibroblast growth factor homologous factors 4 (FHF4) that binds to the C-terminal tail of sodium channels. ZL0177 not only decreases the transient and resurgent currents of Nav1.8 and Nav1.9 but also reduces the proportion of the neurons that generate Nav1.8 resurgent currents. We further show that ZL0177 greatly attenuates the hyperexcitability of small DRG neurons induced by SCI. Taken together, our results indicate Nav1.8 dysfunction following SCI plays a critical role in enhancing excitability of nociceptive neurons. Furthermore, the binding site of FHFs at the C-terminal tail of Nav1.8 and Nav1.9 can serve as a promising therapeutic target for the treatment of SCI pain.

(Received 21 November 2024; accepted after revision 24 July 2025; first published online 22 August 2025)

**Corresponding author** Theodore R. Cummins: Department of Biology, School of Science, Indiana University-Indianapolis, Indianapolis, Indiana 46202, USA. Email: trcummin@iu.edu

### Key points

- Traumatic spinal cord injury (SCI) often leads to chronic pain conditions.
- Recent clinical and experimental studies indicate that the pain induced by SCI may be attributed to abnormal peripheral neuron inputs; however, the molecular mechanisms underlying these changes remain unclear.
- Here we studied sodium currents and action potential firing in rat small DRG neurons following contusive SCI.
- Contusive SCI significantly increased Nav1.8 transient and resurgent sodium currents in the rat small DRG neurons.
- ZL0177, a peptidomimetic inhibitor of fibroblast growth factor homologous factor (FHF) binding to the C-terminus of sodium channels not only decreased Nav1.8/Nav1.9 currents, but also greatly attenuated the hyperexcitability of small DRG neurons induced by SCI. Thus, targeting FHF modulation of Nav1.8/Nav1.9 may serve as a promising treatment strategy for SCI pain.

## Introduction

Traumatic spinal cord injury (SCI) not only results in motor impairment below the injury site but also leads to chronic pain that can last for years. It is reported that up to 86% of individuals with SCI experience clinically significant pain at some time following the injury (Bresnahan et al., 2022; Teasell et al., 2010). Recent clinical and experimental studies indicate that the pain induced by SCI may be attributed to abnormal peripheral inputs. In animal models with contusive injury resembling human SCI, a significant proportion (75%) of small lumbar (L4) dorsal root ganglion (DRG) neurons exhibit spontaneous action potential firing, in contrast to less than 5% of control neurons (Bedi et al., 2010). In addition, SCI alters the functioning of various ion channels (such as Kv3.4, TRPV1, T-type calcium channels, and voltage-gated sodium channels (VGSCs)) in small DRG neurons (Lauzadis et al., 2020; Ritter et al., 2015; Wu et al., 2013; Yang et al., 2014), which serve as primary nociceptors responsible for transmitting pain signals from peripheral nerve terminals to spinal cord neurons.

DRG neurons express at least five VGSC subtypes: Nav1.1, Nav1.6, Nav1.7, Nav1.8, and Nav1.9 (Cummins et al., 2007). These subtypes are typically classified as tetrodotoxin-sensitive (TTX-S; Nav1.1, Nav1.6, and Nav1.7) and tetrodotoxin-resistant (TTX-R; Nav1.8 and Nav1.9). While Nav1.1 and Nav1.6 are predominantly expressed in medium- to large-diameter neurons, Nav1.7 through Nav1.9 are primarily found in small-diameter neurons, where they play distinct roles in the initiation and upstroke phase of action potentials (Black et al., 1996). Yoshimura and de Groat (1997) observed that SCI led to a significant decrease in TTX-R sodium currents and an increase in TTX-S sodium currents in DRG neurons that innervate the bladder. Conversely, Yang et al. (2014) demonstrated an upregulation of Nav1.8 protein in DRG neurons following SCI, and selective knockdown of Nav1.8 suppressed the SCI-induced spontaneous neuronal activity. Consequently, the involvement of VGSCs in

SCI-induced pain remains a topic of debate. Furthermore, in addition to the classic transient sodium currents ($I_{NaT}$) that contribute to action potential depolarization, VGSCs can generate atypical resurgent currents ($I_{NaR}$) under specific conditions during repolarization. Although $I_{NaR}$ are relatively small, they play a critical role in sustaining a high frequency of action potential firing in neurons (Raman & Bean, 1997; Tan et al., 2014). Nav$\beta$4 and A-type fibroblast growth factor homologous factors (FHFs) have been implicated as two major entities of mediating $I_{NaR}$ generation in DRG neurons (Barbosa et al., 2015; Xiao et al., 2022; Xie et al., 2016). Many functional and genetic studies have indicated that an increase in $I_{NaR}$ produced by Nav1.7 and Nav1.8 can induce hyperexcitability in small DRG neurons, leading to conditions such as paroxysmal extreme pain disorder and small fibre neuropathy (Jarecki et al., 2010; Xiao et al., 2019). However, the impact of SCI on $I_{NaR}$ remains unknown.

In this study, we report that contusive SCI causes significant dysregulation of TTX-R sodium currents ($I_{NaT}$ and $I_{NaR}$) and alters the excitability of small DRG neurons in rats. Our findings show that SCI primarily enhances the $I_{NaT}$ and $I_{NaR}$ generated by the Nav1.8 channel. We also observe a notable increase in the proportion of small DRG neurons capable of producing Nav1.8 $I_{NaR}$ following SCI. Current-clamp recordings reveal a marked elevation in the firing frequency of action potentials in small DRG neurons, in the presence of TTX blocking TTX-S VGSCs. We demonstrate that ZL0177, a small peptidomimetic that mimics the critical residues in FHF4 binding to the C-terminal tail of VGSCs (Liu et al., 2019), can reverse the SCI-induced dysfunction of Nav1.8 $I_{NaT}$ and $I_{NaR}$ and significantly reduce the hyperexcitability in SCI-affected DRG neurons. These results strongly suggest that Nav1.8 plays a crucial role in the hyperexcitability of nociceptive neurons, contributing to SCI-induced pain and that the binding site of fibroblast growth factor homologous factors at Nav1.8 C-terminal tail can be a promising therapeutic target for the treatment of SCI pain.

## Methods

### Ethical approval

All procedures were performed in accordance with the Guidelines of the Institutional Animal Care and Use Committee (IACUC) of the Indiana University School of Medicine (approved animal protocol no.: 21098 MD/R/MSS/HZ). All surgical interventions, treatments, and postoperative animal care were also performed in accordance with the *Guide for the Care and Use of Laboratory Animals* (National Research Council). The rats were all individually housed and maintained on a 12 h/12 h light/dark cycle with food and water freely available. The work presented in this study complied with the journal's policies regarding animal experiments.

### Animals and contusive SCI

Female rats are used in experimental SCI research due to the ease of post-operative urinary bladder care. Female Sprague-Dawley rats (210–230 g) were purchased from Envigo (Indianapolis, IN, USA) and were randomly divided into the designated groups. The contusive SCI in rats was performed at the 10th thoracic (T10) vertebral level using an Infinite Horizon Impactor (Infinite Horizons, Lexington, KY, USA) at an impact force of 1.75 newtons for a 0-s dwell time according to our published work (Liu et al., 2022; Liu et al., 2004). Briefly, rats were anaesthetized intraperitoneally with a ketamine (40 mg/kg)/xylazine (5 mg/kg) cocktail, and a laminectomy was carried out at the T9–T10 level. After the exposed vertebral column was stabilized by a vertebral stabilizer, the exposed dorsal surface of the cord was subjected to an impact of 1.75 newtons. After the injury, the muscles and skin were closed in layers, and rats were placed in a temperature- and humidity-controlled chamber overnight. Manual bladder expression was carried out at least three times daily until reflex bladder emptying was established. All rats received a single subcutaneous injection of Ethiqa-XR (0.65 mg/kg) immediately after SCI. Ethiqa-XR is a sustained-release formulation of buprenorphine that provides analgesia for up to 72 h. Animals were closely monitored for signs of pain or distress. If any signs of pain were observed after the initial 72-h period, an additional dose of Ethiqa-XR was administered based on veterinary recommendation. This analgesic protocol was approved by our IACUC. For the sham-operated controls, the animals underwent a T10 laminectomy without the impact. For the naive controls, the animals received no surgery. The animals were killed at 2 weeks after injury for cell culture.

### DRG neuron culture

The lumbar DRG (L1-L6) neurons were acutely dissociated and cultured according to the procedure described previously (Xiao et al., 2019). Briefly, the SCI, sham and naive rats were killed by decapitation after the animals were anaesthetized with intraperitoneal injection of a ketamine (87.7 mg/kg)/xylazine (12.3 mg/kg) cocktail. L1-L6 DRGs were removed quickly from the spinal cord and then incubated in Dulbecco's modified Eagle's medium (DMEM) containing collagenase type I (1 mg/ml; Cat. No.: LS004194, Worthington, Lakewood, NJ, USA) and neutral protease (1 mg/ml; Cat. No.: LS02104, Worthington). The ganglia were sequentially triturated in DMEM supplemented with 10% fetal bovine

serum (FBS) and seeded on glass coverslips coated with poly-D-lysine and laminin. Cultures were maintained at 37°C in a 5% $CO_2$ incubator for up to 48 h.

### ND7/23 cell culture, plasmid and transfection

ND7/23 cells were grown under standard tissue culture conditions (5% $CO_2$ and 37°C) in DMEM supplemented with 10% FBS. The pCMV6-AC-GFP plasmid encoding human FHF4A was purchased from Origene USA Technologies, Inc (Rockville, MD, USA). The cDNA construct encoding human Nav1.8 was subcloned into a pcDNA3.1 expression vector. Using the Invitrogen Lipofectamine 2000, Nav1.8 was transiently co-transfected with FHF4A. The lipofectamine-DNA mixture was added to the cell culture medium and left for 3 h, after which the cells were washed with fresh medium. Cells with green fluorescent protein fluorescence were selected for whole-cell patch-clamp recordings 36–72 h after transfection. ND7/23 cells do not express endogenous TTX-R sodium currents (Nav1.8 or Nav1.9) but do express TTX-S sodium currents (Xiao et al., 2022). Therefore, transfected ND7/23 cells were pretreated with 500 nM TTX to isolate Nav1.8 currents.

### Electrophysiological recordings

Whole-cell voltage-clamp and current-clamp recordings were performed at room temperature (∼21°C) using an EPC-10 amplifier and the Pulse program (HEKA Electronics). The recordings were completed within 36 h of neuron isolation. The DRG neurons that had a membrane capacitance less than 35 pF were patched.

For voltage-clamp recordings, fire-polished electrodes (1.0–2.0 MΩ) were fabricated from 1.7 mm capillary glass using a P-1000 puller (Sutter Instruments), and the tips were coated with sticky wax (KerrLab) to reduce electrode capacitance and enable increased series resistance compensation. The pipette solution contained (in mM): 140 CsF, 1.1 EGTA, 10 NaCl, and 10 HEPES, pH 7.3. The bathing solution contained (in mM): 130 NaCl, 30 TEA chloride, 1 $MgCl_2$, 3 KCl, 1 $CaCl_2$, 0.05 $CdCl_2$, 10 HEPES, and 10 D-glucose, pH 7.3 (adjusted with NaOH). TTX (1 μM) was added to the bath solution to block TTX-S currents in DRG neurons. The liquid junction potential for these solutions was <8 mV; data were not corrected to account for this offset. The offset potential was zeroed before contacting the cell. After establishing whole-cell recording configuration, the resting potential was held at −100 mV for 3 min to allow adequate equilibration between the micropipette solution and the cell interior. Linear leak subtraction, based on resistance estimates from 4 to 5 hyperpolarizing pulses applied before the depolarizing test potential, was

used for all voltage-clamp recordings. Membrane currents were usually filtered at 5 kHz and sampled at 20 kHz. Voltage errors were minimized using 70%–90% series resistance compensation, and the capacitance artefact was cancelled using the computer-controlled circuitry of the patch-clamp amplifier. Nav1.9-like currents elicited by a 50-ms depolarization of −60 mV were measured at 3 min after establishing the whole-cell voltage-clamp mode.

For current-clamp recordings, fire-polished electrodes (3.0–5.0 MΩ) were fabricated from 1.2 mm capillary glass using a P-1000 (Sutter Instruments). The pipette solution contained the following (in mM): 140 KCl, 5 $MgCl_2$, 5 EGTA, 2.5 $CaCl_2$, 4 ATP, 0.3 GTP, and 10 HEPES, pH 7.3 (adjusted with KOH). The bathing solution contained the following (in mM): 140 NaCl, 1 $MgCl_2$, 5 KCl, 2 $CaCl_2$, 10 HEPES, and 10 D-glucose, pH 7.3 (adjusted with NaOH). Neurons were allowed to stabilize for 3 min in the current-clamp mode before initiating current injections to measure action potential activity. TTX (1 μM) was added to the bath solution.

### ZL0177

The small peptidomimetic ZL0177 was synthesized by Biopeptide Co. (San Diego, CA, USA). ZL0177 was dissolved in dimethyl sulfoxide (DMSO) to a stock concentration of 10 mM and stored at −20°C. Stock solution was diluted to the different concentrations of interest in DMEM or bath solution. The final solution contained 0.5% DMSO or compound solution. Cells were pre-incubated in DMEM with vehicle or ZL0177 for 1 h before patch-clamp recording.

### Statistical analysis

Data were analysed using the software programs PulseFit (HEKA) and GraphPad Prism 10 (GraphPad Software, Inc, San Diego, CA, USA). All data are shown as mean ± SD. The number of separate experimental cells was presented as *n*. Between 6 and 16 cells were recorded from each animal. Statistical analysis was performed by Student's *t* test or one-way ANOVA with a *post hoc* test for multiple comparisons. In Figs 5*B* and 9*B*, comparisons were carried out at the same injected current. Comparisons of incidence were performed with Fisher's exact test. $P < 0.05$ indicated a significant difference.

## Results

### SCI increases TTX-R $I_{NaT}$ in small DRG neurons

We first examined whether SCI affects TTX-S and TTX-R $I_{NaT}$ in small DRG neurons. As illustrated in Fig. 1*A*, 1 μM TTX is applied to isolate TTX-R $I_{NaT}$ from the total

currents by blocking the TTX-S component. Compared to the naive and sham groups, SCI results in an increased average density of TTX-R currents in small DRG neurons (Fig. 1*B*). DRG neurons show expression of two TTX-R VGSC subtypes, Nav1.8 and Nav1.9, which can exhibit distinct voltage dependence of activation (Dib-hajj et al., 2002). It is widely recognized that the currents elicited at $-60$ mV and $-5$ mV are predominantly carried by Nav1.9 and Nav1.8, respectively. In Fig. 1*B* and *D*, the current density at $-5$ mV is 43.8% higher in SCI-DRG neurons ($-1.38 \pm 0.14$ nA/pF) than in naive- ($-0.96 \pm 0.09$ nA/pF; *vs.* SCI, $P = 0.0354$) and sham-DRG neurons ($-0.78 \pm 0.10$ nA/pF; *vs.* SCI, $P = 0.0006$). Nav1.9 current evoked at $-60$ mV displays ultra-slow

inactivation and is detectable in 92.3% of naive cells (36/39), 87.5% of sham cells (42/48) and 91.8% of SCI cells (45/49) ($P = 0.8327$), respectively. The current density was not significantly different among the three groups (naive: $-0.07 \pm 0.11$ nA/pF; sham: $-0.07 \pm 0.15$ nA/pF, *vs.* naive $P = 0.6336$; SCI: $-0.07 \pm 0.03$ nA/pF, *vs.* naive $P = 0.9824$; Fig. 1*E*), suggesting that SCI does not affect Nav1.9 current.

TTX-S $I_{NaT}$ are obtained by subtracting the TTX-R component from the total currents (Fig. 1*A*). The TTX-S $I_{NaT}$, probably generated by Nav1.7 with lesser contributions from Nav1.1 and Nav1.6, peaks at approximately $-10$ mV in small DRG neurons (Fig. 1*C*). Compared to the naive group, neither the sham nor the

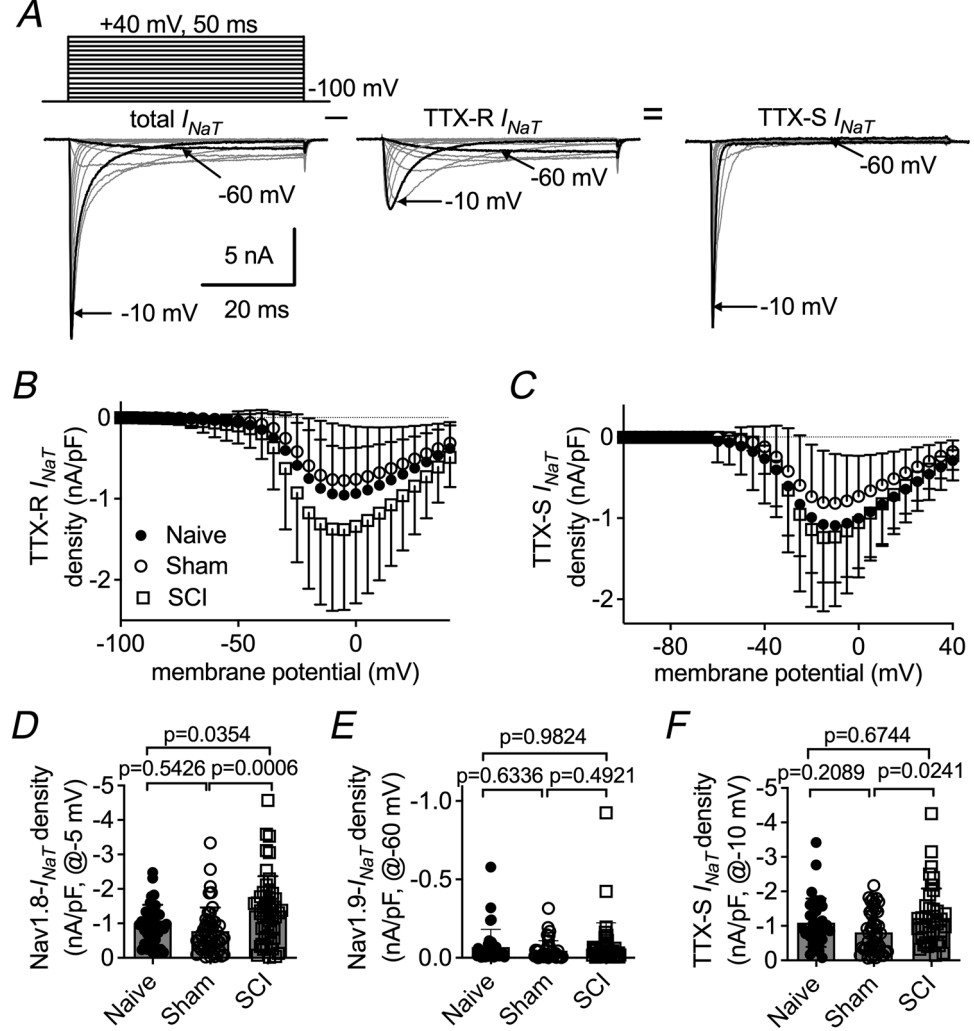

**Figure 1. SCI increases $I_{NaT}$ density in small DRG neurons**
*A*, typical traces of total, TTX-R and TTX-S $I_{NaT}$. $I_{NaT}$ are elicited by 50-ms depolarizing steps to voltages ranging from $-100$ to 40 mV in 10-mV increments (inset). Cells are held at $-100$ mV. TTX: 1 μM. *B*, TTX-R $I_{NaT}$ densities for naive ($n = 38$), sham ($n = 45$) and SCI ($n = 49$) groups. *C*, TTX-S $I_{NaT}$ densities for naive ($n = 36$), sham ($n = 45$) and SCI ($n = 41$) groups. *D*, summary of Nav1.8 $I_{NaT}$ densities measured at $-5$ mV. *E*, summary of Nav1.9 $I_{NaT}$ densities at $-60$ mV. *F*, summary of TTX-S $I_{NaT}$ densities at $-10$ mV. In *D–F*, one-way ANOVA followed by Tukey's *post hoc* test for multiple comparisons was used. $P < 0.05$ indicated a significant change.

SCI surgery significantly alters TTX-S $I_{NaT}$ density (naive: −1.09 ± 0.12 nA/pF; sham: −0.82 ± 0.09 nA/pF, *vs.* naive $P$ = 0.2089; SCI: −1.23 ± 0.13 nA/pF, *vs.* naive $P$ = 0.6744; Fig. 1*F*). However, SCI increases TTX-S $I_{NaT}$ density by 50% compared to the sham group ($P$ = 0.0241; Fig. 1*F*). The voltage dependence of activation and inactivation of TTX-R and TTX-S sodium channels remains unchanged among the three groups: the curves of normalized steady-state activation (Fig. 2*A*, *C*) and inactivation (Fig. 2 *B*, *D*) nearly completely overlap each other.

### SCI affects TTX-R $I_{NaR}$ generation in small DRG neurons

We next examined whether SCI affects the $I_{NaR}$ generated by TTX-R and TTX-S VGSCs in small DRG neurons. Unlike $I_{NaT}$, $I_{NaR}$ is typically elicited by a repolarizing step to an intermediate voltage following a depolarizing step

to +30 mV. As shown in Fig. 3*A*, an endogenous $I_{NaR}$ with slow onset and extremely slow decay kinetics can be elicited in small DRG neurons from naive, sham and SCI groups in the presence of 1 μM TTX. The TTX-R $I_{NaR}$ is observed in a subpopulation (25/54 cells, 46%) of DRG neurons isolated from naive rats (Fig. 3*B*). The percentage of neurons exhibiting TTX-R $I_{NaR}$ remains unchanged in the sham group (29/60 cells, 48%; $P$ = 0.8529). However, this percentage roughly doubles following SCI (52/55 cells, 94%; $P$ < 0.0001; Fig. 3*C*). In naive DRG neurons, TTX-R $I_{NaR}$ can be elicited at voltages ranging from −50 to +20 mV, peaking at approximately −20 mV with a density of −9.6 ± 1.3 pA/pF, which is 0.8% ± 0.1% of the peak $I_{NaT}$ (Fig. 3*D*, *E*). SCI surgery increases the density and the $I_{NaR}$-to-$I_{NaT}$ ratio to −18.3 ± 1.6 pA/pF (*vs.* naive, $P$ = 0.0007) and 1.3% ± 0.1% (*vs.* naive, $P$ = 0.0010), respectively, but the sham procedure does not significantly alter the density (−9.5 ± 1.2 pA/pF; *vs.* naive, $P$ = 0.9993) or the ratio of TTX-R $I_{NaR}$ (0.8% ± 0.1%; *vs.* naive,

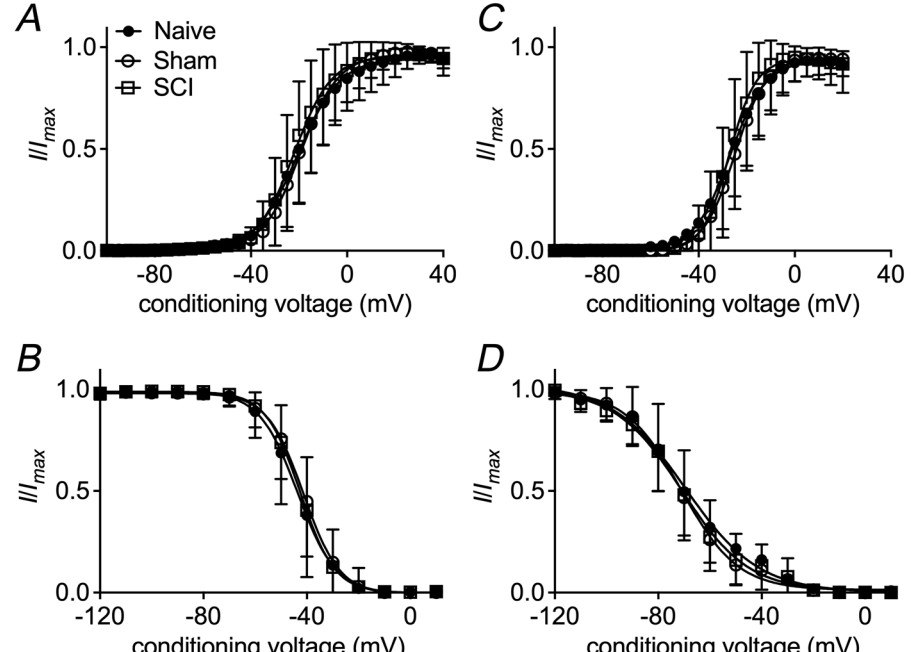

**Figure 2. SCI does not alter gating properties of TTX-R (*A*, *B*) or TTX-S (*C*, *D*) sodium channels in small DRG neurons**

The voltage dependence of steady-state activation is estimated based on the data from Fig. 1. The channel conductance is calculated using the equation: $G(Nav) = I/(V − V_{rev})$ in which $I$, $V$, and $V_{rev}$ represent inward current value, membrane potential, and reversal potential, respectively. Data are plotted as a fraction of the maximum conductance. The voltage dependence of steady-state inactivation is estimated with a standard double-pulse protocol in which sodium currents are induced by a 20-ms depolarizing potential of 0 mV following a 500-ms pre-pulse at voltages ranging from −120 to +10 mV. Currents are plotted as a fraction of the maximum peak current. Data points are fitted with the Boltzmann equation. $V_{1/2}$ for activation of TTX-R VGSCs: naive, −20.5 ± 0.4 mV, $n$ = 38; sham, −19.4 ± 0.3 mV, $n$ = 48; SCI, −22.1 ± 0.3 mV, $n$ = 49. $V_{1/2}$ for activation of TTX-S VGSCs: naive, −26.9 ± 0.5 mV, $n$ = 36; sham, −25.0 ± 0.3 mV, $n$ = 45; SCI, −27.3 ± 0.3 mV, $n$ = 41. $V_{1/2}$ for steady-state inactivation of TTX-R VGSCs: naive, −43.6 ± 0.5 mV, $n$ = 38; sham, −41.4 ± 0.4 mV, $n$ = 39; SCI, −42.6 ± 0.4 mV, $n$ = 42. $V_{1/2}$ for steady-state inactivation of TTX-S VGSCs: naive, −69.5 ± 1.1 mV, $n$ = 32; sham, −71.1 ± 0.6 mV, $n$ = 43; SCI, −70.9 ± 0.9 mV, $n$ = 36.

$P = 0.9283$). Neither the sham procedure nor SCI surgery modifies the voltage dependence of TTX-R $I_{NaR}$ activation (Fig. 3$D$, $E$).

TTX-S $I_{NaR}$ are derived by subtracting the TTX-R component from the total $I_{NaR}$, which are elicited by 100-ms hyperpolarizing steps to potentials ranging from $+20$ to $-100$ mV. In contrast to the slow TTX-R component, TTX-S $I_{NaR}$ exhibit fast kinetics, characterized by a rapid onset and decay (Fig. 4$A$). These currents are observed only in a small subpopulation (3/46 cells, 6.5%) of small naive DRG neurons (Fig. 4$B$ and $C$). This is consistent with our previous observation that TTX-S $I_{NaR}$ are primarily generated by large DRG

neurons (Cummins et al., 2005). TTX-S $I_{NaR}$ can be induced at voltages ranging from $-70$ to $-10$ mV, peaking at approximately $-35$ mV with a density of $-6.7 \pm 1.3$ pA/pF (Fig. 4$D$), which represents 0.6% $\pm$ 0.3% of the peak $I_{NaT}$ elicited at $-10$ mV. SCI surgery does not change the density of TTX-S $I_{NaR}$ ($-8.9 \pm 2.2$ pA/pF; *vs.* naive, $P = 0.6371$) but increases the ratio (1.5% $\pm$ 0.3%, *vs.* naive, $P = 0.0394$; Fig. 4$E$). Sham surgery changes neither the density ($-9.9 \pm 1.8$ pA/pF; *vs.* naive, $P = 0.4042$) nor the ratio (0.8% $\pm$ 0.2%; *vs.* naive, $P = 0.6897$). None of the surgical interventions significantly alters the proportion of the neurons exhibiting TTX-S $I_{NaR}$ (sham: 4/45 cells, *vs.* naive, $P = 0.7139$; SCI, 4/45 cells, *vs.* naive,

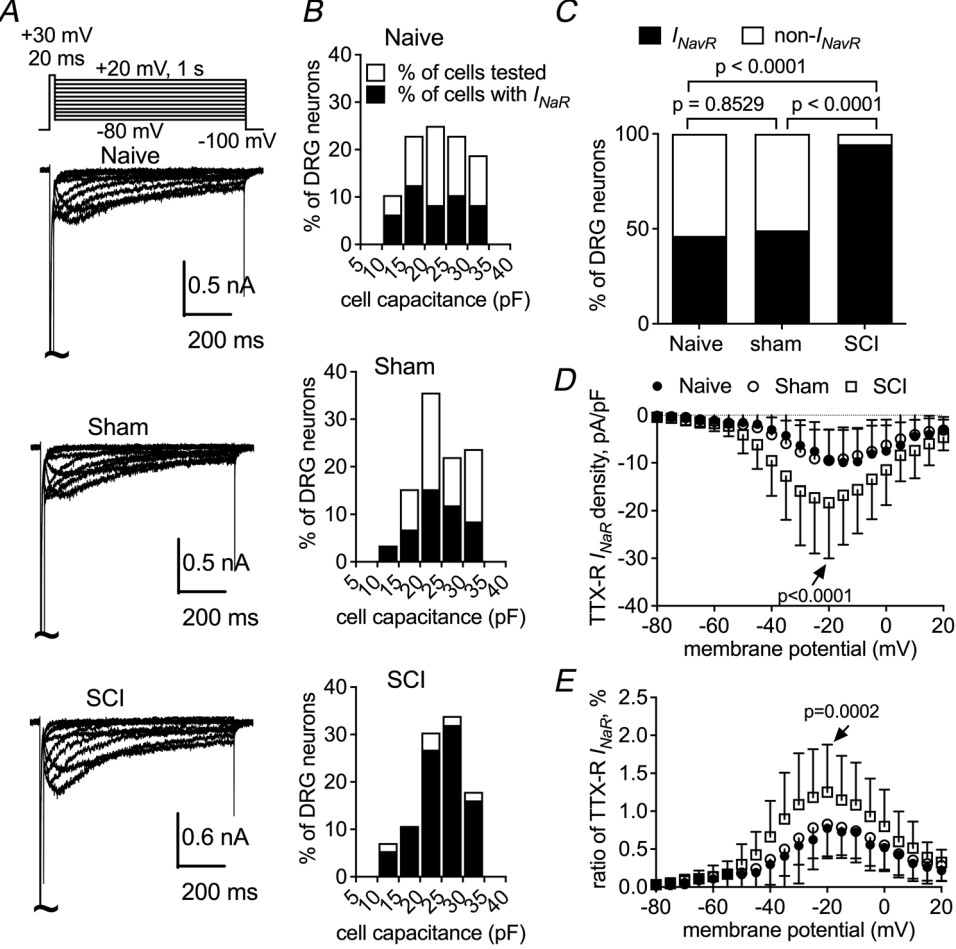

**Figure 3. SCI significantly increases slow TTX-R $I_{NaR}$ in small DRG neurons**
*A*, typical traces of slow TTX-R $I_{NaR}$ in naive, sham and SCI DRG neurons. $I_{NaR}$ are elicited by a two-step protocol, in which the cell membrane is initially depolarized to $+30$ mV for 20 ms, followed by a 1-s hyperpolarizing steps to potentials ranging from $+20$ to $-80$ mV (inset). *B*, the frequency of DRG neurons with the capability of generating slow TTX-R $I_{NaR}$. Open bar: the frequency of DRG neurons tested. Filled bar: the frequency of DRG neurons producing TTX-R $I_{NaR}$. *C*, comparison of the percentage of DRG neurons with TTX-R $I_{NaR}$ under naive, sham and SCI. Fisher's exact test was used. *D*, TTX-R $I_{NaR}$ density TTX-R $I_{NaR}$ are normalized to the capacitance of DRG neurons. *E*, voltage dependence of the relative TTX-R $I_{NaR}$. TTX-R $I_{NaR}$ are normalized to the peak transient current elicited at 0 mV. Cells are treated with 1 μM TTX. In *D* and *E*: naive, $n = 25$; sham, $n = 29$; SCI, $n = 52$; one-way ANOVA followed by Tukey's *post hoc* test for multiple comparisons was used. $P < 0.05$ indicated a significant change.

*P* = 0.7139; Fig. 4*C*) or the voltage dependence of TTX-S $I_{NaR}$ activation (Fig. 4*B–D*).

## TTX-R VGSC dysfunction contributes to hyperexcitability of small SCI-DRG neurons

We further investigated whether the dysfunction of TTX-R VGSCs alters the excitability of small DRG neurons when TTX-S VGSCs are blocked. It is known that nociceptive DRG neurons may generate action potentials under these conditions (Renganathan et al., 2001; Tan et al., 2014). As illustrated in Fig. 5*A* and *B*, SCI significantly increases the frequency of action potential firing in small DRG neurons when a 2-s injection of 300 pA is applied. The number of evoked action potentials

is 1.0 ± 1.3 for the naive group, 1.0 ± 0.7 for the sham group, and 7.6 ± 16.5 for the SCI group (*vs.* naive, *P* = 0.0333; *vs.* sham, *P* = 0.0386). A similar increasing trend is observed with current injections greater than 300 pA (Fig. 5*B*). SCI does not alter the resting membrane potential (Fig. 5*C*) or the threshold current for initiating action potentials (Fig. 5*D*). Moreover, SCI does not affect the percentage of neurons exhibiting spontaneous firing (Fig. 5*E*).

## ZL0177 decreases Nav1.8 $I_{NaT}$ and $I_{NaR}$ in a heterologous system

We next determined whether the hyperexcitability of small SCI-DRG neurons could be reversed by decreasing

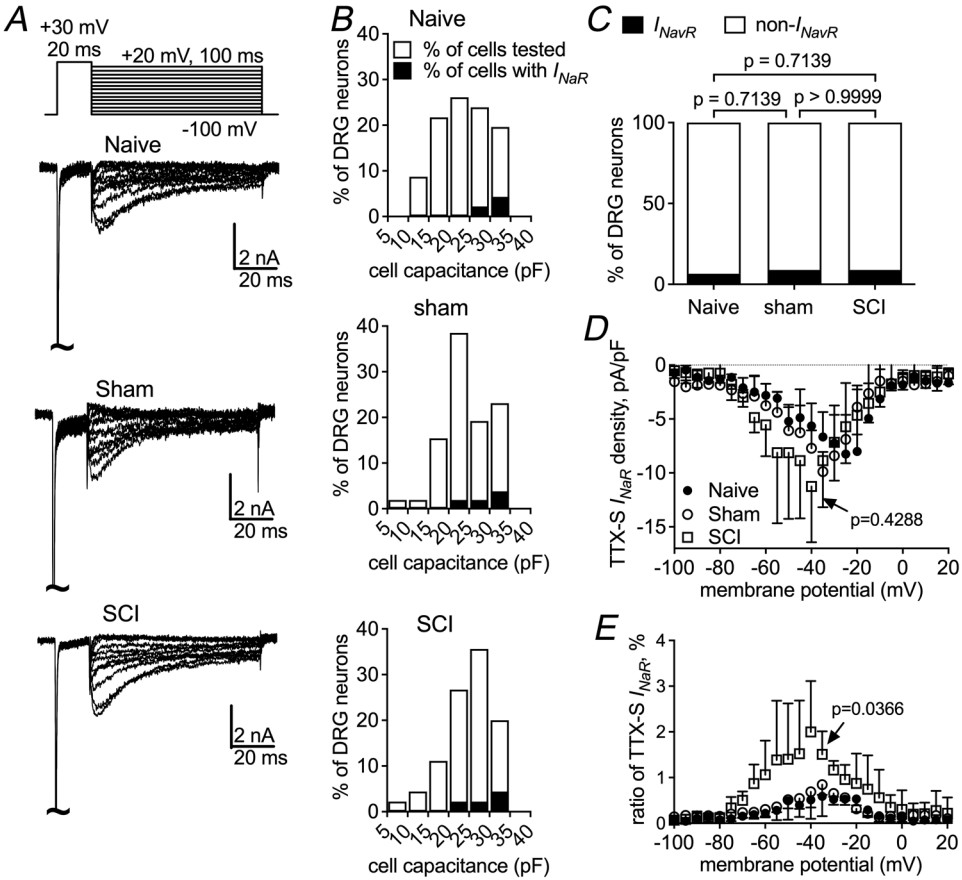

**Figure 4. SCI does not significantly affect TTX-S $I_{NaR}$ in small DRG neurons**
*A*, typical traces of TTX-S $I_{NaR}$ elicited in naive, sham and SCI DRG neurons. $I_{NaR}$ are elicited by a two-step protocol, in which cell membrane is initially depolarized to +30 mV for 20 ms, followed by a 100-ms hyperpolarizing steps to potentials ranging from +20 to −100 mV (inset). TTX-S $I_{NaR}$ are obtained by subtracting the traces recorded after 1 μM TTX treatment from the total $I_{NaR}$ traces. *B*, the frequency of DRG neurons displaying the capability of generating TTX-S $I_{NaR}$. Open bar: the frequency of DRG neurons tested. Filled bar: the frequency of DRG neurons producing TTX-S $I_{NaR}$. *C*, comparison of the percentage of DRG neurons with TTX-S $I_{NaR}$ under naive, sham and SCI. Fisher exact test was used. *D*, TTX-S $I_{NaR}$ density. TTX-S $I_{NaR}$ are normalized to the capacitance of DRG neurons. *E*, voltage dependence of activation of the relative TTX-S $I_{NaR}$. TTX-S $I_{NaR}$ are normalized to the peak transient current elicited at −10 mV. In *D* and *E*: naive, *n* = 3; sham, *n* = 4; SCI, *n* = 4; one-way ANOVA followed by Tukey's *post hoc* test for multiple comparisons was used. *P* < 0.05 indicated a significant change.

Nav1.8 $I_{NaT}$ and $I_{NaR}$. To achieve this goal, we first tested whether ZL0177 influences FHF4A-mediated Nav1.8 $I_{NaR}$, given that FHF4A has been implicated as a critical mediator of Nav1.8 $I_{NaR}$ in primary neurons (Xiao et al., 2022). Importantly, the $I_{NaR}$ can be fully reconstituted in a heterologous system by co-expressing recombinant Nav1.8 and FHF4A (Fig. 6A and B). Liu et al. (2019) reported that ZL0177 disrupts the interaction between Nav1.6 and FHF4. As shown in Fig. 6C and D, ZL0177 simultaneously inhibits Nav1.8 $I_{NaT}$ and $I_{NaR}$ in ND7/23 cells in a concentration-dependent manner. The $IC_{50}$ values estimated are 19.5 μM and 25.3 μM, respectively. At 30 μM, ZL0177 inhibits 60.2% of $I_{NaT}$ and reduces 50.0% of the ratio of $I_{NaR}$ to the peak $I_{NaT}$ from 5.8% ± 0.5% to 2.9% ± 0.2%.

−60 mV by approximately 75%, respectively (Fig. 7A). ZL0177 positively shifts the voltage dependence of channel activation by 6.1 mV (DMSO: −16.8 ± 0.6 mV vs. ZL0177: −10.7 ± 0.5 mV, P = 0.0255; Fig. 7B). However, it does not affect steady-state inactivation (DMSO: −36.9 ± 0.9 mV vs. ZL0177: −34.8 ± 0.7 mV, P = 0.7291; Fig. 7C) or the recovery rate from inactivation (Fig. 7D). The compound may preferentially influence the generation of TTX-R $I_{NaR}$. It not only reduces the percentage of neurons exhibiting TTX-R $I_{NaR}$ from 86% (18/21 cells) to 24% (6/25 cells; P < 0.0001; Fig. 8A) but also decreases the ratio of TTX-R $I_{NaR}$, elicited at −20 mV, from 1.4% ± 0.3% to 0.3% ± 0.1% (P < 0.0001) (Fig. 8B). ZL0177 does not significantly alter the voltage dependence of TTX-R $I_{NaR}$ activation (Fig. 8B).

### ZL0177 inhibits TTX-R $I_{NaT}$ and $I_{NaR}$ in small SCI-DRG neurons

In small SCI-DRG neurons, 30 μM ZL0177 inhibits TTX-R $I_{NaT}$ at −5 mV by approximately 58% and at

### ZL0177 suppresses hyperexcitability of small SCI-DRG neurons

ZL0177 can reverse the hyperexcitability of small DRG neurons observed in the presence of TTX following

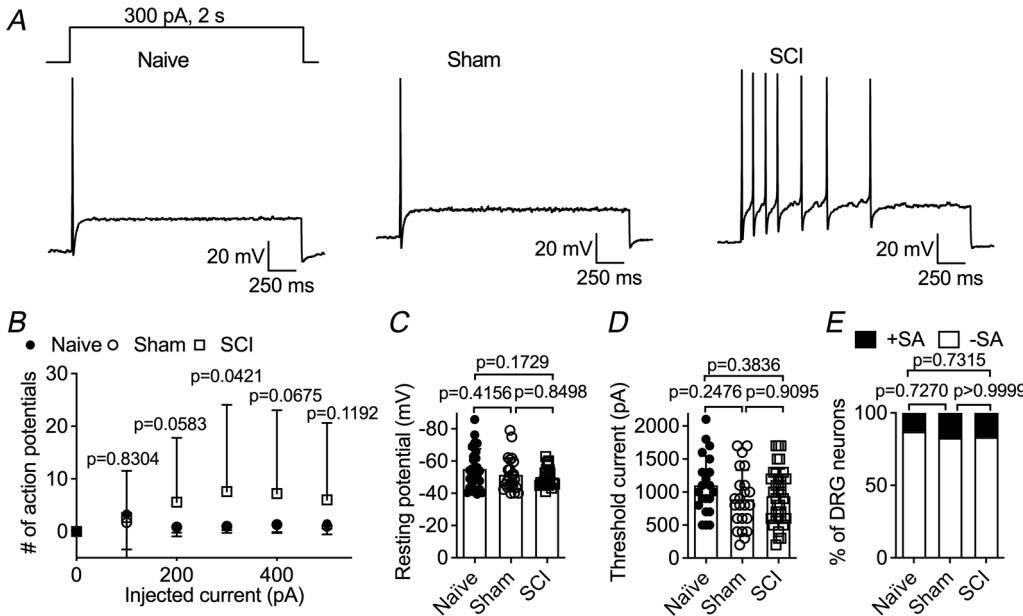

**Figure 5. SCI upregulates excitability of small DRG neurons**
*A*, typical action potential firing elicited by a 2-s injection of 300 pA (inset) in small DRG neurons from naive, sham and SCI rats. *B*, summary of the number of action potentials elicited by a 2-s injection of currents ranging from 0 to 500 pA. At 300 pA, the number of evoked action potentials is 1.0 ± 1.3 for the naive group (*n* = 22), 1.0 ± 0.7 for the sham group (*n* = 20), and 7.6 ± 16.5 for the SCI group (*n* = 34). One-way ANOVA followed by Benjamini and Yekutieli's *post hoc* test for multiple comparisons was used. *C*, comparison of the resting membrane potentials. Naive, *n* = 22; sham, *n* = 20; SCI, *n* = 35. The resting membrane potentials are −55.1 ± 12.5 mV for the naive group, −51.5 ± 10.4 mV for the sham group and −49.9 ± 5.9 mV for the SCI group, respectively (naive *vs.* sham, *P* = 0.2669; naive *vs.* SCI, *P* = 0.0632; sham vs. SCI, *P* = 0.5183). The neurons with resting membrane potentials >−40 mV were excluded. *D*, comparison of the threshold currents required for initiation of action potentials. Current threshold was measured by a 1-ms injection of step current, which ranged from 0 to 3000 pA in 100-pA increment steps with an interval of 5 s. *E*, comparison of the proportion of DRG neurons with spontaneous firing. SA: spontaneous active. In *C–E*, one-way ANOVA followed by Tukey's *post hoc* test for multiple comparisons was used. *P* < 0.05 indicated a significant change.

SCI. In the presence of 30 µM ZL0177, the number of action potentials evoked by a 2-s injection of 300 pA decreases from $9.9 \pm 12.8$ (DMSO) to $1.3 \pm 1.0$ (ZL0177; $P = 0.0136$; Fig. 9*A*, *B*). This decrease is also observed when the injected current ranges from 100 to 400 pA. ZL0177 does not alter the resting membrane potential (DMSO, $-40.8 \pm 4.6$ mV *vs.* ZL0177, $-42.4 \pm 3.1$ mV, $P = 0.2593$; Fig. 9*C*), but it greatly increases the threshold current required to initiate action potential firing (DMSO, $888 \pm 438$ pA *vs.* ZL0177, $1592 \pm 562$ pA, $P = 0.0012$; Fig. 9*D*). It is worth noting that the resting membrane potential measured here is less than the value shown in Fig. 5*D*. This might be caused by the concentration (0.5%) of DMSO that can increase leak current under whole-cell recording configuration.

## Discussion

We demonstrate that, in a widely accepted *in vivo* SCI model, contusive injury at the T10 vertebra results in hyperexcitability of L1-L6 nociceptive neurons along with an increased density of TTX-R $I_{NaT}$, enhanced generation of TTX-R $I_{NaR}$, and a greater percentage of neurons producing TTX-R $I_{NaR}$. Our findings suggest that the dysfunctional upregulation of Nav1.8 currents is a major contributor to the hyperexcitability of nociceptive neurons induced by SCI.

Our study reveals that SCI primarily modulates the density of the sodium currents flowing through the Nav1.8 subtype. This finding aligns with previous research that indicated significant increase of Nav1.8 protein in L4 and L5 DRG neurons after SCI (Yang et al., 2014). Moreover, increased expression of Nav1.8 has also been observed in the rat models of bone cancer pain, scorpion sting-induced pain and chronic peripheral inflammatory pain (Belkouch et al., 2014; Liu et al., 2014; Ye et al., 2016). Nav1.8 is responsible for most of the sodium current that contributes to the upstroke of action potentials in nociceptive neurons. Computer simulations demonstrate that an increase in Nav1.8 conductance can elevate the firing frequency of action potentials in these neurons (Choi & Waxman, 2011). Supporting this conclusion, our data indicate that in small DRG neurons, when TTX-S VGSCs are blocked, the firing frequency is significantly higher in the SCI group compared to the naive and sham groups. In addition to dysregulation of

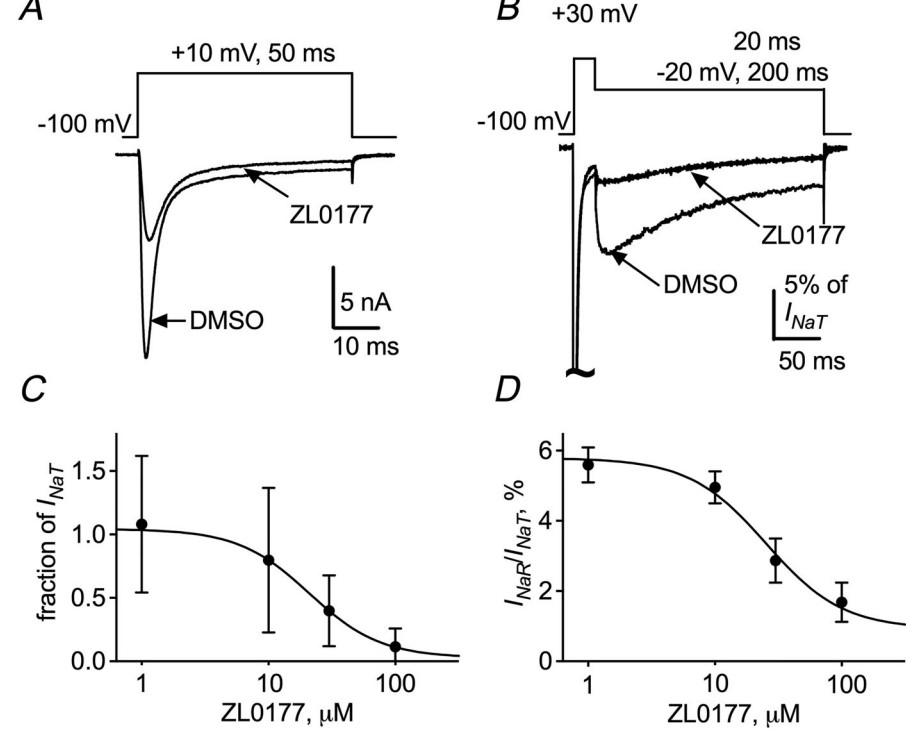

**Figure 6. ZL0177 inhibits Nav1.8 $I_{NaT}$ and $I_{NaR}$ in ND7/23 cells co-transfected with FHF4A**
*A*, typical Nav1.8 $I_{NaT}$ traces elicited by a 50-ms depolarization of +10 mV (inset) in the absence and presence of 30 µM ZL0177. *B*, typical Nav1.8 $I_{NaR}$ traces evoked by a 200-ms repolarization of −20 mV, following a 20-ms depolarization of +30 mV (inset), in the absence and presence of 30 µM ZL0177. *C*, concentration-dependent inhibition by ZL0177 of $I_{NaT}$. $I_{NaT}$ is normalized to the average amplitude of the current measured under vehicle (0.5% DMSO). *D*, concentration-dependent inhibition by ZL0177 of the ratio of FHF4A-mediated Nav1.8 $I_{NaR}$. Cells are held at −100 mV. Each data point, shown as mean ± SD, comes from 5–14 experimental cells.

Nav1.8 expression, subtle alternations in Nav1.8 gating properties, for example a ∼−6 mV shift in channel activation or a +7 mV shift in fast inactivation, could also lead to hyperexcitability of nociceptive neurons (Faber et al., 2012; Han et al., 2014, 2018). However, in our study SCI does not significantly alter the midpoint ($V_{1/2}$) of activation or steady-state inactivation of either TTX-S or TTX-R VGSCs (Fig. 2).

In addition to Nav1.8, small DRG neurons also exhibit high expression levels of other VGSC subtypes, such as Nav1.7 and Nav1.9, all of which play a vital role in determining the excitability of DRG neurons. Unlike Nav1.7 and Nav1.8, Nav1.9 produces a slowly inactivating and persistent TTX-R current that activated at hyper-polarized potentials. While Nav1.8 contributes to most of the rising phase of action potentials, Nav1.7 and Nav1.9 are likely to play important roles in modulating

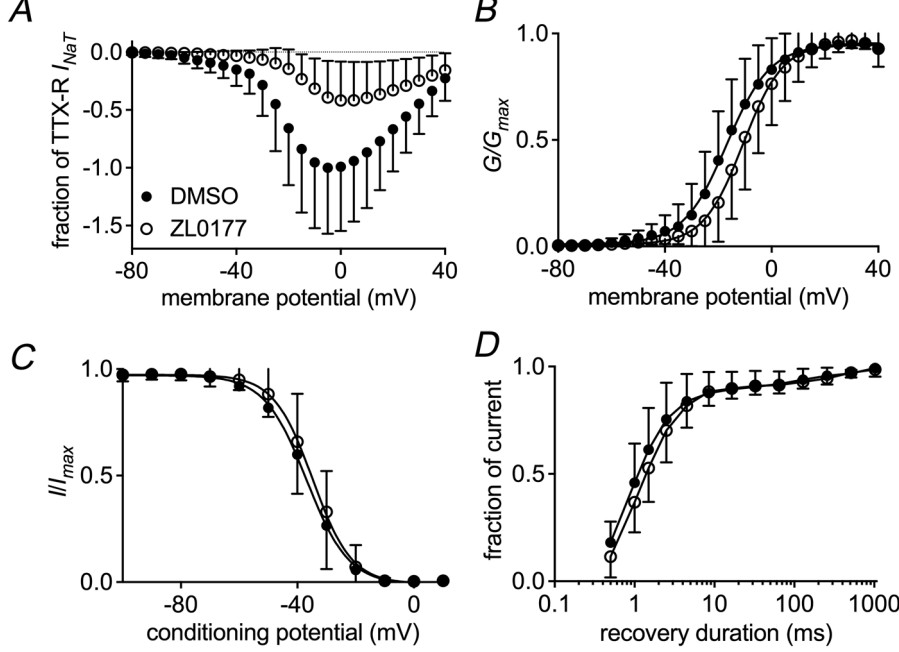

**Figure 7. Effects of 30 μM ZL0177 on TTX-R sodium currents in small SCI-DRG neurons**
*A*, effect of ZL0177 on TTX-R $I_{NaT}$. Families of TTX-R sodium currents are elicited as described in Fig. 1. The currents are normalized to the average amplitude of the peak transient current measured at −5 mV in the presence of vehicle (DMSO). *B*, effect of ZL0177 on voltage dependence of channel activation. *C*, effect of ZL0177 on steady-state inactivation. Steady-state inactivation is assayed as described in Fig. 2. *D*, effect of ZL0177 on recovery rate from inactivation. Recovery from inactivation is assayed by a protocol in which the cells are prepulsed to 0 mV for 50 ms to inactivate sodium channels and then brought back to −100 mV for increasing recovery durations before the test pulse to 0 mV. Student's *t* test was used. *P* < 0.05 indicated a significant change.

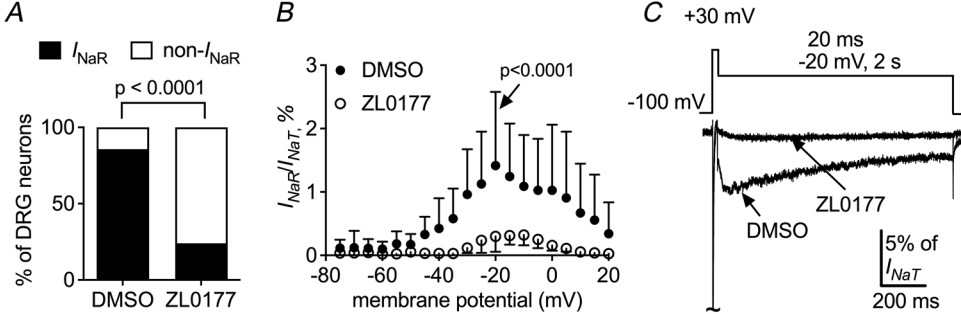

**Figure 8. Effects of 30 μM ZL0177 on TTX-R $I_{NaR}$ in small SCI-DRG neurons**
*A*, effect of ZL0177 on the percentage of the neurons with TTX-R $I_{NaR}$. Fisher's exact test was used. *B*, effect of ZL0177 on the ratio of TTX-R $I_{NaR}$. TTX-R $I_{NaR}$ are elicited as described in Fig. 3. DMSO, *n* = 18; ZL0177, *n* = 6. Student's *t* test was used. *C*, TTX-R $I_{NaR}$ traces were elicited by a 1000-ms repolarization of −20 mV, following a 20-ms depolarization of +30 mV (inset), in the absence and presence of 30 μM ZL0177. *P* < 0.05 indicated a significant change.

subthreshold potential and resting membrane potential, respectively (Cummins et al., 2007). Our study reveals that, compared to the naive and sham groups, SCI shows an increasing trend in the current densities of TTX-S VGSCs. Interestingly, in contrast to our findings, Yoshimura and de Groat (1997) reported that the spinal transection at T8-T9 decreases TTX-R $I_{NaT}$ and increases TTX-S $I_{NaT}$ in DRG neurons that innervate the hypertrophic bladder. These findings suggest that the type of spinal cord injury may exert divergent influences on the regulation of VGSC subtype expression in DRG neurons.

We observed a significant increase in the endogenous TTX-R $I_{NavR}$, which we attribute to Nav1.8. In our previous work, both Nav1.8 and Nav1.9 TTX-R VGSCs demonstrated an intrinsic ability to generate $I_{NaR}$ in heterologous systems and primary neurons (Xiao et al., 2022). The $I_{NaR}$ flowing through Nav1.8 and Nav1.9 exhibit distinct kinetics. For instance, Nav1.8 $I_{NaR}$ peaks at −20 mV and displays slow onset and extremely slow decay kinetics, while Nav1.9 $I_{NaR}$ peaks at −85 mV and exhibits fast onset and fast decay kinetics, resembling the currents produced by TTX-S VGSCs. The endogenous TTX-R $I_{NavR}$ reported here is similar to the currents

generated by recombinant Nav1.8 transiently transfected into DRG neurons, showing similar onset and decay kinetics, as well as voltage dependence of activation (Xiao et al., 2019). Nav1.9 $I_{NaR}$ was probably not detected in this study for several reasons: (1) Nav1.9 generates $I_{NaR}$ only in approximately 30% of small DRG neurons (Xiao et al., 2022); (2) Nav1.9 currents run down rapidly under whole-cell recording conditions (TTX-R sodium currents are recorded after the total currents are elicited and are treated with TTX); and (3) Nav1.9 $I_{NaR}$ may be masked by robust Nav1.8 $I_{NaR}$, as nociceptor neurons frequently express both Nav1.8 and Nav1.9.

We note that in this study we focused on small DRG neurons based on previous studies implicating them in SCI-induced neuropathic pain (Bedi et al., 2010). In this study, we choose small DRG neurons below 35 pF in capacitance, which correlates to 37.5 μm in diameter. These small DRG neurons are often assumed to be unmyelinated C-fibre neurons and play important roles in conveying the nociceptive, thermal and mechanoreceptive signals. Recent transcriptomic studies have created a high-resolution map of DRG neuron subtypes and have shown that the majority of the small DRG

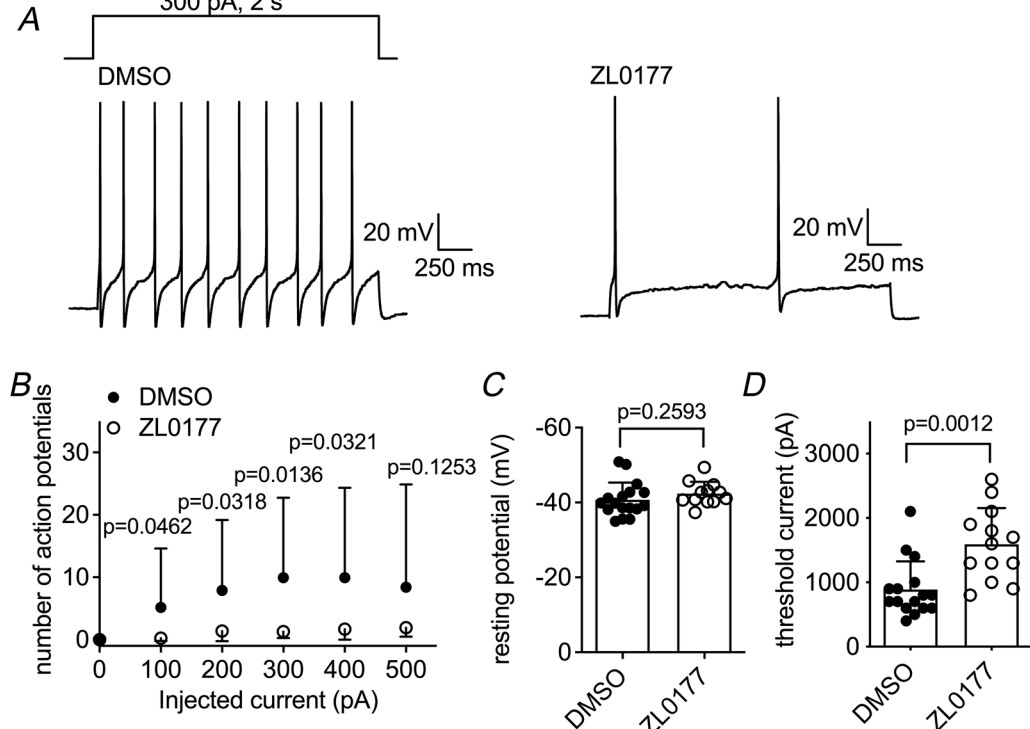

**Figure 9. ZL0177 partially reverses hyperexcitability of small SCI-DRG neurons**
*A*, typical action potential firing is elicited by a 2-s injection of 300 pA (inset) in the presence of vehicle (0.5% DMSO; left) and 30 μM ZL0177 (right). *B*, effects of ZL0177 ($n = 13$) on firing frequency of action potentials. DMSO, $n = 17$. *C*, effects of ZL0177 ($n = 12$) on resting membrane potential. DMSO, $n = 17$. Neurons with resting membrane potentials >−35 mV were excluded. *D*, effects of ZL0177 ($n = 13$) on threshold current. DMSO, $n = 16$. Current threshold was measured as described in the legend of Fig. 5*D*. In *B–D*, Student's *t* test was used. $P < 0.05$ indicated a significant change.

neurons belong to the cluster of nociceptive neurons. They express nociceptor markers such as TRPA1 or Nav1.8 (Tavares-Ferreira et al., 2022). Consistent with this finding, it has been demonstrated that most of the small DRG neurons response to capsaicin and bind isolectin B4 (Odem et al., 2018). Our electrophysiological recordings also show that nearly all small DRG neurons (naive: 37/38; sham: 46/48; SCI: 48/49) can produce sodium currents generated by Nav1.8 or Nav1.9, two proteins critical for pain sensations (Cummins et al., 2007). Although the majority of DRG neurons tested here are likely to be nociceptive neurons, transcriptomic analysis has identified multiple C-fibre nociceptor subtypes in rodent and human (Tavares-Ferreira et al., 2022) DRG neurons. It would be interesting to know if Nav1.8 $I_{NaR}$ is predominantly expressed in specific subtypes under control conditions and which subtypes exhibit increased Nav1.8 $I_{NaR}$ after SCI using patch-seq experiments (Caldwell et al., 2016).

As a result of our focus on small neurons, we did not capture data on larger DRG neuron populations, which can include A$\beta$ nociceptors. Medium to large DRG neurons express substantially larger TTX-S $I_{NaR}$ than small DRG neurons (Cummins et al., 2005). While these larger neurons have not been directly implicated in SCI-induced neuropathic pain, the TTX-S $I_{NaR}$ in larger diameter neurons has been associated with mechanical hypersensitivity and neuronal hyperexcitability in a rat model of radicular pain induced by local dorsal root ganglion inflammation (Xie et al., 2016). Therefore, based on the data that we present here, we cannot rule out the possibility that TTX-S $I_{NaR}$ are also altered in larger DRG neurons following SCI.

Our data demonstrate for the first time that aberrant Nav1.8 $I_{NaR}$ may contribute to the hyperexcitability of nociceptive neurons associated with SCI-induced neuropathic pain. SCI not only doubles Nav1.8 $I_{NaR}$ amplitude but also doubles the percentage of small DRG neurons capable of producing Nav1.8 $I_{NaR}$. Although $I_{NaR}$ are small, they can play a crucial role in sustaining a high frequency of action potential firing in Purkinje neurons (Khaliq et al., 2003). In our previous works, increased Nav1.8 $I_{NaR}$ significantly upregulated action potential firing frequency in DRG neurons (Tan et al., 2014). Conversely, a reduction in Nav1.8 $I_{NaR}$ leads to downregulation of neuronal excitability (Xiao et al., 2019). Increased Nav1.8 $I_{NaR}$ is also implicated as a critical factor contributing to DRG neuron excitability associated with small fibre neuropathy. Therefore, we propose that both increased Nav1.8 $I_{NaR}$ and $I_{NaT}$ are essential contributors to the hyperexcitability of nociceptive neurons following SCI.

However, in contrast to previous findings showing that the proportion of small DRG neurons with spontaneous firing substantially increases after SCI (Bedi et al., 2010; Wu et al., 2013; Yang et al. 2014), we did not observe

a change in spontaneous firing. This discrepancy may reflect two important differences: first, we concentrate on TTX-R VGSCs and measured excitability of DRG neurons in the presence of TTX, which blocks all TTX-S VGSCs including Nav1.7, which can amplify subthreshold depolarizations; second, we extracted the DRG neurons at 2 weeks, not 4 weeks, after SCI surgery. Dysregulation of other ion channels or proteins that contribute to spontaneous activity in neurons at 4 weeks may not be evident at 2 weeks.

Our data provide the first direct evidence that molecular manipulation of Nav1.8 $I_{NaR}$ can be achieved in nociceptive neurons by targeting the FHF binding site in the C-terminal tail of VGSCs. A-type FHFs (FHF1A–FHF4A) and FHF2B are expressed throughout peripheral sensory neurons. The four A-type isoforms each consist of a divergent N-terminal tail, a conserved $\beta$-trefoil core, and a short C-terminal tail. The long N-terminal tail, particularly the amino acid segment located at the very beginning, acts as an open-channel blocker of VGSCs and is responsible for the generation of TTX-R $I_{NaR}$ (Xiao et al., 2022). ZL0177 disrupts the interaction between FHF4A and Nav1.6 by docking at the binding site of FHF4 $\beta$-trefoil domain in the VGSC C-terminal tail (Liu et al., 2019). As we predicted, ZL0177 inhibits TTX-R $I_{NaR}$ in our heterologous system co-expressing recombinant FHF4A and Nav1.8, as shown in Fig. 6. Since it seems unlikely that ZL0177 directly targets the FHF4A N-terminal tail, the inhibition of Nav1.8 $I_{NaR}$ most likely results from ZL0177 disrupting the interaction between the FHF4A $\beta$-trefoil domain and the Nav1.8 C-terminal tail. The unbinding of $\beta$-trefoil domain from the cytoplasmic tail of VGSCs (Goetz et al., 2009; Liu et al., 2001) is expected to substantially decrease the local concentration of the N-terminus near the channel pore. Additionally, ZL0177 positively shifts the voltage dependence of Nav1.8 activation, thereby reducing the 'window currents' region, where the channels activate but do not fully inactivate. Lewis and Raman (2013) showed that open-channel blockers might have lower affinity in VGSCs with DIVS4 in the resting or partially deployed configuration compared to when DIVS4 is fully deployed. FHF2B resembles the A-type FHF $\beta$-trefoil domain but lacks the long N-terminal tail. Recently, we demonstrated that FHF2B can facilitate the generation of Nav1.8 $I_{NaR}$ induced by exogenous peptides (Xiao et al., 2024). We conclude that ZL0177 preferentially depresses Nav1.8 $I_{NaR}$ in DRG neurons by inhibiting the interaction of either A-type FHFs or FHF2B with the Nav1.8 C-terminal tail. Interestingly, the Laezza group recently identified another compound (1028) that modulates the assembly of the FGF14/Nav1.6 complex (Dvorak et al., 2025). In contrast to ZL0177, compound 1028 directly binds to FGF14 but not the Nav1.6 C-terminal tail. Without affecting current density

or channel activation, compound 1028 positively shifted steady-state inactivation of Nav1.6. It will be interesting to investigate if compound 1028 alters generation of Nav1.8 $I_{NaR}$ in sensory neurons.

We also observed a significant reduction in TTX-R $I_{NaT}$ fluxing through Nav1.8 and Nav1.9 in small DRG neurons pretreated with ZL0177. Liu et al. (2019) reported a similar reduction of recombinant Nav1.6 $I_{NaT}$ in a heterologous system. This reduction may be attributed to ZL0177's inhibition of VGSC trafficking, which can be impacted by FHFs. ZL0177 forms interactions with Asp1833, His1843, Arg1866, Phe1873, Tyr1883, Arg1891 and Arg1892 in Nav1.6. Sequence alignment indicates that these residues are highly conserved among Nav1.6, Nav1.8 and Nav1.9, suggesting that ZL0177 may also disrupt the interaction of FHF4A and Nav1.8 as well as Nav1.9. Another possibility is that ZL0177 itself can inhibit VGSCs by modulating the gating properties of the channels. A-type FHFs and FHF2B are potent Nav1.8 gating modifiers; they can negatively shift voltage dependence of Nav1.8 activation, positively shift voltage dependence of steady-state inactivation, and accelerate recovery rate from inactivation. However, these effects induced by FHFs are not completely reversed in the presence of ZL0177. As shown in Fig. 7, ZL0177 shifts the voltage dependence of the TTX-R sodium channel activation to more positive potentials in DRG neurons without affecting steady-state inactivation or the rate of recovery from inactivation. Interestingly, ZL0177-induced inhibition of TTX-R $I_{NaT}$ and $I_{NaR}$ reverses the hyperexcitability of small SCI-DRG neurons. This reversal may reflect actions on both Nav1.8 and Nav1.9.

Specific inhibition of Nav1.8 has been shown to be effective in moderate-to-severe acute pain (McCoun et al., 2025). Previous work has shown that Nav1.8-specific knockdown suppresses the hyperexcitability of DRG neurons following SCI (Yang et al., 2014). Given the critical role of Nav1.8 in SCI-induced neuropathic pain, we further propose that the FHF binding site in the C-terminal tail of Nav1.8 may present a promising target for developing novel analgesics to treat SCI-related pain.

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

## Additional information

### Data availability statement

The data supporting the results of this study are included within the figures in the published article and are available from the corresponding author upon reasonable request.

### Competing interests

The authors declare that they do not have any competing interests.

### Author contributions

Y.X., Y.P., N.L. and T.R.C. conceived and conducted the project and wrote the manuscript. Y.X., Y.P. and T.R.C. contributed to the acquisition, analysis or interpretation of data. Y.X. performed neuron isolation and whole-cell voltage-clamp recordings. Y.P. performed voltage-clamp and current-clamp recordings. N.L. performed sham and contusive spinal cord injury surgery. All authors contributed to and approved the final version of the manuscript. All persons designated as authors qualify for authorship, and all those who qualify for authorship are listed.

### Funding

This work was supported by the Indiana Spinal Cord & Brain Injury Research Fund from the Indiana State Department of Health (2020) (Y.X.), Craig H Neilsen Foundation 882060 (Y.X. and N.L.) and R01NS053422 (T.R.C.).

### Keywords

DRG neurons, pain, resurgent sodium currents, sodium channels, spinal cord injury

## Supporting information

Additional supporting information can be found online in the Supporting Information section at the end of the HTML view of the article. Supporting information files available:

**Peer Review History**

