## [Peer Review History · The Journal of Physiology]

Contusive spinal cord injury causes Nav1.8 dysfunction to upregulate small sensory neuron excitability

Yucheng Xiao, Yanling Pan, Naikiu Liu, and Theodore R Cummins

DOI: 10.1113/JP288215

Corresponding author(s): Theodore Cummins (trcummin@iu.edu)

The following individual(s) involved in review of this submission have agreed to reveal their identity: Edgar Walters (Referee #2)

Review Timeline:

Submission Date:	21-Nov-2024
Editorial Decision:	24-Jan-2025
Revision Received:	28-Apr-2025
Editorial Decision:	30-May-2025
Revision Received:	19-Jun-2025
Editorial Decision:	08-Jul-2025
Revision Received:	22-Jul-2025
Accepted:	24-Jul-2025

Senior Editor: Nathan Schoppa

Reviewing Editor: Carole Torsney

Transaction Report:

Dear Dr Cummins,

Re: JP-RP-2024-288215 "Contusive spinal cord injury causes Nav1.8 dysfunction to upregulate small sensory neuron excitability" by Yucheng Xiao, Yanling Pan, Naikiu Liu, and Theodore R Cummins

Thank you for submitting your manuscript to The Journal of Physiology. It has been assessed by a Reviewing Editor and by 2 expert referees and we are pleased to tell you that it is potentially acceptable for publication following satisfactory major revision.

REVISION CHECKLIST:

Please upload two versions of your manuscript text: one with all relevant changes highlighted and one clean version with no

changes tracked. The manuscript file should include all tables and figure legends, but each figure/graph should be uploaded as separate, high-resolution files.

We look forward to receiving your revised submission.

Yours sincerely,

Nathan Schoppa
Senior Editor
The Journal of Physiology

REQUIRED ITEMS

- Author photo and profile. First or joint first authors are asked to provide a short biography (no more than 100 words for one author or 150 words in total for joint first authors) and a portrait photograph. These should be uploaded and clearly labelled together in a Word document with the revised version of the manuscript. See Information for Authors for further details.

- You must start the Methods section with a paragraph headed Ethical approval (https://jp.msubmit.net/cgi-bin/main.plex?form_type=display_requirements#methods).

Research must comply with The Journal's policies regarding animal experiments (<https://physoc.onlinelibrary.wiley.com/hub/animal-experiments>) and adherence to these policies must be stated in the manuscript.

Authors should confirm in their Methods section that their experiments were carried out according to the guidelines laid down by their institution's animal welfare committee, including an ethics approval reference number. The Methods section must contain a statement about access to food, water and housing, details of the anaesthetic regime: anaesthetic used, dose and route of administration, and method of killing the experimental animals.

- Your manuscript must include a complete Additional Information section, including competing interests; funding; author contributions and acknowledgements.

- Please upload separate high-quality figure files via the submission form.

- Please ensure that any tables are editable and in Word format, and wherever possible, embedded in the article file itself.

- Please ensure that the Article File you upload is a Word file.

- Papers must comply with the Statistics Policy: https://jp.msubmit.net/cgi-bin/main.plex?form_type=display_requirements#statistics.

In summary:

- If n {less than or equal to} 30, all data points must be plotted in the figure in a way that reveals their range and distribution. A bar graph with data points overlaid, a box and whisker plot or a violin plot (preferably with data points included) are acceptable formats.
- If $n > 30$, then the entire raw dataset must be made available either as supporting information, or hosted on a not-for-profit repository, e.g. FigShare, with access details provided in the manuscript.
- 'n' clearly defined (e.g. x cells from y slices in z animals) in the Methods. Authors should be mindful of pseudoreplication.
- All relevant 'n' values must be clearly stated in the main text, figures and tables.
- The most appropriate summary statistic (e.g. mean or median and standard deviation) must be used. Standard Error of the Mean (SEM) alone is not permitted.
- Exact p values must be stated. Authors must not use 'greater than' or 'less than'. Exact p values must be stated to three significant figures even when 'no statistical significance' is claimed.

- Please include an Abstract Figure file, as well as the Figure Legend text within the main article file. The Abstract Figure is a piece of artwork designed to give readers an immediate understanding of the research and should summarise the main conclusions. If possible, the image should be easily 'readable' from left to right or top to bottom. It should show the physiological relevance of the manuscript so readers can assess the importance and content of its findings. Abstract Figures should not merely recapitulate other figures in the manuscript. Please try to keep the diagram as simple as possible and without superfluous information that may distract from the main conclusion(s). Abstract Figures must be provided by authors no later than the revised manuscript stage and should be uploaded as a separate file during online submission labelled as File Type 'Abstract Figure'. Please also ensure that you include the figure legend in the main article file. All Abstract Figures should be created using BioRender. Authors should use The Journal's premium BioRender account to export high-resolution images. Details on how to use and access the premium account are included as part of this email.

- Please include a full title page as part of your main article (Word) file, which should contain the following: title, authors, affiliations, corresponding author name and contact details, keywords, and running title.

Reviewing Editor:

Ethics Concerns:

I don't believe anesthesia monitoring is described?
Euthanasia anesthesia not described?

Comments to ensure the paper complies with the Statistics Policy:

Please note:

- If $n \leq 30$: All data points must be plotted to display range and distribution.
- A bar graph with data points overlaid, a box and whisker plot or a violin plot (the latter two also preferably with data points included) are acceptable formats.
- Note: if each subject has numerous data points associated with it (e.g. time course data), treat 'n' as being each data point, not the number of subjects.
- If $n > 30$: Data points do not need be plotted in the figure but the entire raw dataset must be available as supporting information or hosted in a not for profit repository

NB Exact p values must be stated to three significant figures.

Data summaries should be presented as mean (SD). SEM should not be used.

Comments to the authors:

This ms by Xiao et al. that explores the mechanism underlying small sensory neuron hyperexcitability following SCI has been reviewed by 2 reviewers. The reviewers raise a number of concerns that must be fully addressed.

Senior Editor:

Comments to ensure the paper complies with the Statistics Policy:

Please indicate SD rather than SEM.

If $n < 30$, please show points for individual experiments in the plots.

Please provide precise p-values in Figure legends as well as the main text.

Comments to the authors:

Thank you for submitting your manuscript to Journal of Physiology. It has been reviewed by two expert reviewers, who were both quite positive about the potential impact of the study. They have however brought up several concerns, all of which will need to be addressed. The points raised should be addressable with changes in the text and some additional analyses, not requiring more experiments. For example, both reviewers requested more information about the identification and categorization of DRG neurons. Between the reviewers and editors, we have also noted the following points around animal care and statistics that require being addressed:

(1) Around animal care, please be sure to provide information about animal access to food and water, monitoring of

anesthesia during surgeries, and the anesthesia used during euthanasia.

(2) Around statistics, please indicate SD rather than SEM. Also, if $n < 30$, please show points for individual experiments in the plots. Please also provide precise p-values in Figure legends as well as the main text. We appreciate that they appear in the main text, but having them the legend will help readers.

Referee #1:

This study from the Cummins group investigated the effects of spinal cord injury (SCI) in a rat contusion model on small dorsal root ganglion (DRG) neurons, focusing on changes in sodium currents. They observed increased transient and resurgent sodium currents, primarily through TTXr channels in small DRG neurons after SCI, which they attribute to Nav1.8. Using ZL0177, a small peptidomimetic targeting the C-terminal tail of sodium channels, they tested its ability to reduce these currents and the hyperexcitability of DRG neurons. Their experiments included analyzing the effects of ZL0177 on Nav1.8 currents and its influence on SCI-induced neuronal changes. In their study, they claim that the inhibitor reduced Nav1.8 currents and the excitability of the DRGs.

The study is well performed and described, and the data are worth publishing. Investigations into TTXr resurgent currents are not very often performed and here a link to SCI induced hyperexcitability may be established. The SCI induced hyperexcitability is only visible upon current injection of 300pA - the only readout for hyperexcitability was increase of APs of tonically firing DRG neurons and determining threshold.

Major:

- The authors should distinguish better between Nav1.8 and Nav1.9 mediated resurgent currents. Please show example traces in Figs. 1 for currents elicited at -5mV and at -60mV. Please single out Nav1.9, and show Nav1.9 specific resurgent currents. Line 188: how many of the patched cells displayed Nav1.9? Fig. 2: Please distinguish between Nav1.8 and Nav1.9, or at least discuss the contribution of Nav1.9. Fig. 8: please show current traces.

Line 381f: "Nav1.9 generates INaR only in a small fraction of small DRG

382 neurons" - please show and analyze.

Line 273-274: "There is an upward trend in the current density elicited at -60 mV; however, this increase is not statistically significant among the three groups (naïve: 0.02 {plus minus} 0.01 nA/pF; sham: -0.02 {plus minus} 0.01 nA/pF; SCI: -0.07 {plus minus} 0.03 nA/pF; $p = 0.0807$; Fig. 1E), suggesting that SCI does not greatly affect Nav1.9 current. " p-values do not provide information about the size of the effect. Was this experiment sufficiently powered to reliably detect an effect at -60 mV (which is smaller than the effect at 5 mV)? I would not be sure to "suggest that SCI does not greatly affect Nav1.9", especially with a p-value of 0.08.

- Subgrouping of DRG neurons: the authors chose size as sole distinction criteria. Recent transcriptomic studies suggest that this may not be sufficient to identify those neurons responsible for neuropathic pain. The authors should explain why they used only size and put their choice into context with recent findings from transcriptomic atlases published on mice, rat and other species (e.g. line 73ff, 370ff). Please discuss how to potentially match your choice of DRG subgrouping with that of transcriptomic data.

- o At least subgrouping into tonic vs phasic firing would help. This may have significant impact on the results shown in Fig. 5 and 9 on SCI induced hyperexcitability and its reversal.

o It may be worth considering additional readouts for hyperexcitability.

o Line 370ff "Taken together, these findings suggest that the type of spinal cord injury may exert divergent influences on the regulation of VGSC subtype expression in DRG neurons." Maybe also sensory neuron subtypes play a role, and a more detailed characterization of the neurons you patched is necessary/helpful.

• The histograms in Figs. 5 and 9 need to be replaced by dot plots.

• Line 345-347/Figure 5/Figure 9: The authors show that SCI does not affect the current threshold but causes hyperexcitability (Figure 5). ZL0177 partially reverses hyperexcitability and increases the current threshold (Figure 9).

Line 345-347: "ZL0177 [...] can decrease both TTX-R INaT and INaR, effectively reversing the hyperexcitability of small DRG neurons following SCI." It seems that ZL0177 does not mitigate the SCI-associated neuronal hyperexcitability through the same mechanism as it is caused by. SCI does not affect current threshold, but ZL0177 does. Even though ZL0177 lowers Nav1.8 current density, the lowered excitability might just as well be caused by effects on Nav1.7 or Nav1.9 (the increased current threshold in combination with no change in the RMP points to Nav1.7 in my opinion). How specific is ZL0177? How well is the C-terminus conserved across Nav1.7/Nav1.8/Nav1.9? Could the effects also be mediated by affecting Nav1.7? A change in threshold suggests effects of ZL on currents active during subthreshold depolarization, not so much currents that support repetitive firing as used as readout for hyperexcitability in this study. Please explain this discrepancy and discuss.

Minor:

• There is a recent paper by Laezza group using FHF4 blockers (<https://pubmed.ncbi.nlm.nih.gov/39747162/>), please discuss and explain the difference in substances used. Also in terms of shifts of activation (line 305ff).

• Line 273-274: "Average firing frequency is 1.0 {plus minus} 0.2 for the naïve group, 1.0 {plus minus} 0.1 for the sham group, and 7.0 {plus minus} 2.6 for the SCI group ($p = 0.0447$; one-way ANOVA)." Unit of frequency is missing.

• Throughout the "results"-section: Sometimes the statistical test is provided after the p-value, sometimes it is not.

• Line 361f: please compare to results from the literature.

Referee #2:

Evidence continues to increase for major contributions of hyperactivity in primary nociceptors to persistent pain in various pain conditions, including central neuropathic pain after spinal cord injury (SCI). However, the mechanisms underlying pain-driving nociceptor hyperexcitability are not well understood. The rigorous, well-designed experiments in this manuscript significantly extend our understanding of the contributions of Nav1.8 to the hyperexcitability of probable nociceptors reported to be critical for SCI-induced enhancement of evoked and spontaneous pain-like behavior in rats after SCI (Yang et al. 2014). The authors provide the first independent confirmation that nociceptor hyperexcitability after SCI involves enhanced Nav1.8 function. Even more important, they have revealed that the enhanced Nav1.8 function involves a substantial increase in resurgent current (TTX-R INaR; which does not occur for the TTX-sensitive currents that are probably conducted through Nav1.7 and Nav1.6 channels) and that nociceptor hyperexcitability after SCI can be suppressed by a drug (ZL0177) that they have shown to inhibit FHF4A-mediated TTX-R INaR. Thus, this study adds significantly to the mechanistic understanding of an important mechanism for SCI pain. Given that nociceptor hyperactivity is being found to contribute to many other pain conditions, these findings are likely to have implications beyond SCI. I have a few suggestions to strengthen the manuscript.

A general suggestion is to discuss more explicitly how the SCI-induced increase in TTX-R INaR may alter DRG neuron function. First, it is worth mentioning why most of the small DRG neurons may be nociceptors, which strengthens the implications of the SCI-induced hyperexcitability for driving pain (e.g., nociceptors are enriched in small DRG neurons and in Nav1.8-expressing DRG neurons, previous studies using similar dissociation conditions found SCI-induced hyperexcitability primarily in capsaicin-sensitive DRG neurons as well as in IB4+ neurons).

Second, is the enhanced TTX-R INaR so brief that it would only be expected to promote relatively high-frequency bursting (e.g., ~5 Hz, as in Fig.9A), or could it contribute to lower-frequency repetitive firing or ongoing activity? In particular, is the decay time constant for TTX-R INaR sufficiently long that it could contribute to the low-frequency nociceptor spontaneous activity (SA, often 0.1-2 Hz) that has been associated with SCI. This would need a comment about why the authors did not find SCI-induced SA under their conditions, which differ in multiple ways from those in prior nociceptor SA studies (e.g., female rather than male rats, 2 weeks vs {greater than or equal to} 4 weeks post-SCI test time, 5 mM rather than 3 mM K⁺ in the extracellular solution, 175 vs 150 Kdyn spinal impact - with the impact for a 0-s (?) dwell time vs 1-s dwell time).

Third, some TTX-R INaR is seen at -50 and -45 mV in Fig. 3D, which is in the range where pharmacological evidence for a partial contribution of Nav1.8 to depolarizing spontaneous fluctuations (DSFs) of membrane potential was reported for mouse DRG neurons (PMID 37862056). It may be plausible that TTX-R INaR also contributes to DSF generation following each AP or burst of APs.

MINOR COMMENTS

Line 63 - It is not self-evident why pain below and at the SCI level suggests abnormal peripheral inputs. Please clarify.

Line 66 - The Siddall 1995 reference is inappropriate - they did not do any experiments on SCI-induced SA in lumbar DRG neurons, none of which were reported until 2010.

Line 125 - Which collagenase and protease were used?

Line 137 - Do ND7/23 express endogenous Nav1.9 current?

Line 173 - The unit of statistical analysis is the individual cell or neuron. How many neurons were recorded from each rat?

Fig. 3C, Fig.8A - It is impossible to have 150% of the neurons express any kind of current. End the Y-axis at 100%.

Fig.6 - It is surprising to have Nav1.8 currents generated by a 10 or 30 mV depolarization from a holding potential of -100 mV. Are these the absolute voltages rather than changes in voltage? If not, a comment about why this occurs in these cells would be helpful.

Line 396 - Add "Purkinje" before "neurons" for clarity.

Line 436 - Correct to Yang et al., 2014.

END OF COMMENTS

Response to Editors and Reviewers:

We appreciate the thoughtful comments from the editors and the reviewers. We have carefully considered all the concerns and suggestions. We modified the manuscript and the figures to address the critiques. Below please find our detailed response to the reviewers (responses in *italics*).

REQUIRED ITEMS

- Author photo and profile. First or joint first authors are asked to provide a short biography (no more than 100 words for one author or 150 words in total for joint first authors) and a portrait photograph. These should be uploaded and clearly labelled together in a Word document with the revised version of the manuscript. See Information for Authors for further details.

The joint first authors' photos and profiles are added. Please see lines 38-44 in the revised manuscript.

- You must start the Methods section with a paragraph headed Ethical approval (https://jp.msubmit.net/cgi-bin/main.plex?form_type=display_requirements#methods).

Now the revised manuscript started the methods section with a paragraph headed Ethical approval. Please see lines 114-120.

Research must comply with The Journal's policies regarding animal experiments (<https://physoc.onlinelibrary.wiley.com/hub/animal-experiments>) and adherence to these policies must be stated in the manuscript.

The statement is included in the revised manuscript. Please see lines 119-120.

Authors should confirm in their Methods section that their experiments were carried out according to the guidelines laid down by their institution's animal welfare committee, including an ethics approval reference number. The Methods section must contain a statement about access to food, water and housing, details of the anaesthetic regime: anaesthetic used, dose and route of administration, and method of killing the experimental animals.

We have modified the methods section according to the above requests. Please see lines 118-119, 137-139.

- Your manuscript must include a complete Additional Information section, including competing interests; funding; author contributions and acknowledgements.

We have included these additional sections. Please see lines 418-434.

- Please upload separate high-quality figure files via the submission form.

We have uploaded the high-quality figure files.

- Please ensure that any tables are editable and in Word format, and wherever possible, embedded in the article file itself.

We do not have any tables in the revised manuscript.

- Please ensure that the Article File you upload is a Word file.

We have uploaded the Word file.

- Papers must comply with the Statistics Policy: https://jp.msubmit.net/cgi-bin/main.plex?form_type=display_requirements#statistics.

We have revised the manuscript to comply with the Statistics Policy.

In summary:

- If n {less than or equal to} 30, all data points must be plotted in the figure in a way that reveals their range and distribution. A bar graph with data points overlaid, a box and whisker plot or a violin plot (preferably with data points included) are acceptable formats.

- If $n > 30$, then the entire raw dataset must be made available either as supporting information, or hosted on a not-for-profit repository, e.g. FigShare, with access details provided in the manuscript.

- 'n' clearly defined (e.g. x cells from y slices in z animals) in the Methods. Authors should be mindful of pseudoreplication.

We have shown all bar graphs with datapoints overlaid and have clearly defined “n”.

- All relevant 'n' values must be clearly stated in the main text, figures and tables.

We have clearly stated “n” throughout the manuscript.

- The most appropriate summary statistic (e.g. mean or median and standard deviation) must be used. Standard Error of the Mean (SEM) alone is not permitted.

We have used mean and standard deviation.

- Exact p values must be stated. Authors must not use 'greater than' or 'less than'. Exact p values must be stated to three significant figures even when 'no statistical significance' is claimed.

We have provided all exact p values.

- Please include an Abstract Figure file, as well as the Figure Legend text within the main article file. The Abstract Figure is a piece of artwork designed to give readers an immediate understanding of the research and should summarise the main conclusions. If possible, the image should be easily 'readable' from left to right or top to bottom. It should show the physiological relevance of the manuscript so readers can assess the importance and content of its findings. Abstract Figures should not merely recapitulate other figures in the manuscript. Please try to keep the diagram as simple as possible and without superfluous information that may distract from the main conclusion(s). Abstract Figures must be provided by authors no later than the revised manuscript stage and should be uploaded as a separate file during online submission labelled as File Type 'Abstract Figure'. Please also ensure that you include the figure legend in the main article file. All Abstract Figures should be created using BioRender. Authors should use The Journal's premium BioRender account to export high-resolution images. Details on how to use and access the premium account are included as part of this email.

We have created the abstract figure using BioRender and uploaded it separately. The abstract figure legend is on lines 63-67.

- Please include a full title page as part of your main article (Word) file, which should contain the following: title, authors, affiliations, corresponding author name and contact details, keywords, and running title.

We have included the full title page.

Reviewing Editor:

Ethics Concerns:

I don't believe anesthesia monitoring is described?

Euthanasia anesthesia not described?

We have added the description of anesthetization in Method (DRG neuron culture). Please see lines 126-128 and 138-139.

Comments to ensure the paper complies with the Statistics Policy:

Please note:

- If $n \leq 30$: All data points must be plotted to display range and distribution.
 - A bar graph with data points overlaid, a box and whisker plot or a violin plot (the latter two also preferably with data points included) are acceptable formats.
 - Note: if each subject has numerous data points associated with it (e.g. time course data), treat 'n' as being each data point, not the number of subjects.
 - If $n > 30$: Data points do not need be plotted in the figure but the entire raw dataset must be available as supporting information or hosted in a not for profit repository
- NB Exact p values must be stated to three significant figures.
Data summaries should be presented as mean (SD). SEM should not be used.

As requested, the bar graphs with datapoints overlaid, "n" as being each data point, exact p values and mean \pm SD are used in the revised manuscript.

Comments to the authors:

This ms by Xiao et al. that explores the mechanism underlying small sensory neuron hyperexcitability following SCI has been reviewed by 2 reviewers. The reviewers raise a number of concerns that must be fully addressed.

Senior Editor:

Comments to ensure the paper complies with the Statistics Policy:

Please indicate SD rather than SEM.

If $n < 30$, please show points for individual experiments in the plots.

Please provide precise p-values in Figure legends as well as the main text.

As requested, the bar graphs with datapoints overlaid, "n" as being each data point, exact p values and mean \pm SD are used in the revised manuscript.

Comments to the authors:

Thank you for submitting your manuscript to Journal of Physiology. It has been reviewed by two expert reviewers, who were both quite positive about the potential impact of the study. They have however brought up several concerns, all of which will need to be addressed. The points raised should be addressable with changes in the text and some additional analyses, not requiring more experiments. For example, both reviewers requested more information about the identification and categorization of DRG neurons. Between the reviewers and editors, we have also noted the following points around animal care

and statistics that require being addressed:

(1) Around animal care, please be sure to provide information about animal access to food and water, monitoring of anesthesia during surgeries, and the anesthesia used during euthanasia.

We have provided the information. Please see lines 118-119, 127 and 137-139.

(2) Around statistics, please indicate SD rather than SEM. Also, if $n < 30$, please show points for individual experiments in the plots. Please also provide precise p-values in Figure legends as well as the main text. We appreciate that they appear in the main text, but having them in the legend will help readers.

We have shown all data as mean \pm SD, the data points for individual experiments in the plots and the precise p-values in Figure legends.

Referee #1:

This study from the Cummins group investigated the effects of spinal cord injury (SCI) in a rat contusion model on small dorsal root ganglion (DRG) neurons, focusing on changes in sodium currents. They observed increased transient and resurgent sodium currents, primarily through TTXr channels in small DRG neurons after SCI, which they attribute to Nav1.8. Using ZL0177, a small peptidomimetic targeting the C-terminal tail of sodium channels, they tested its ability to reduce these currents and the hyperexcitability of DRG neurons. Their experiments included analyzing the effects of ZL0177 on Nav1.8 currents and its influence on SCI-induced neuronal changes. In their study, they claim that the inhibitor reduced Nav1.8 currents and the excitability of the DRGs.

The study is well performed and described, and the data are worth publishing. Investigations into TTXr resurgent currents are not very often performed and here a link to SCI induced hyperexcitability may be established. The SCI induced hyperexcitability is only visible upon current injection of 300pA - the only readout for hyperexcitability was increase of APs of tonically firing DRG neurons and determining threshold.

We thank the reviewer for their supportive comments.

Major:

- The authors should distinguish better between Nav1.8 and Nav1.9 mediated resurgent currents. Please show example traces in Figs. 1 for currents elicited at -5mV and at -60mV. Please single out Nav1.9, and show Nav1.9 specific resurgent currents. Line 188: how many of the patched cells displayed Nav1.9? Fig. 2: Please distinguish between Nav1.8 and Nav1.9, or at least discuss the contribution of Nav1.9. Fig. 8: please show current traces.

Thank you. We have shown the example traces elicited at -10 mV (the increment is 10-mV in Fig. 1A) and at -60 mV. However, it is difficult to show Nav1.9 specific resurgent currents. DRG neurons often co-express Nav1.8 and Nav1.9. The amplitude of Nav1.8 current is typically much larger than that of Nav1.9 current. In our previous work, Nav1.9 specific resurgent currents were only detected in a small fraction (~30%) of Nav1.8-knockdown DRG neurons. In wild type DRG neurons, it is highly possible that Nav1.9 I_{NaR} are already masked by Nav1.8 tail currents or Nav1.8 I_{NaR} . In the Discussion section, we have discussed the probability and proposed that the TTX-R resurgent currents observed in this study were mainly mediated by Nav1.8. Please see lines 319-331.

The fraction of the patched cells, in which Nav1.9 current was elicitable at -60 mV, was 36/39 for Naïve, 42/48 for Sham and 45/49 for SCI, respectively. Analysis of Fisher's exact test indicated that the change

was not statistically significant ($p = 0.8327$). After removing the cells without Nav1.9 currents, one-way ANOVA showed that the change was not statistically significant either among Naïve, Sham and SCI groups ($p = 0.4833$). Please see lines 206-210.

As suggested, we have discussed the difference between Nav1.8 and Nav1.9 and the contribution of Nav1.9 in Discussion section. Please see lines 308-311.

We have shown the current traces in Figure 8.

Line 381f: "Nav1.9 generates INaR only in a small fraction of small DRG neurons" - please show and analyze.

As indicated above, Nav1.9 specific resurgent currents are difficult to determine in wild type DRG neurons. So, we do not measure Nav1.9 resurgent currents in this study. Based on our previous work with Nav1.8-knockdown DRG neurons, only around 30% small DRG neurons expressing Nav1.9 generate Nav1.9 resurgent currents (Xiao et al., 2022). We apologize for the confusion and this crucial reference is now cited at the corresponding place (line 328).

Line 273-274: "There is an upward trend in the current density elicited at -60 mV; however, this increase is not statistically significant among the three groups (naïve: 0.02 ± 0.01 nA/pF; sham: -0.02 ± 0.01 nA/pF; SCI: -0.07 ± 0.03 nA/pF; $p = 0.0807$; Fig. 1E), suggesting that SCI does not greatly affect Nav1.9 current." p-values do not provide information about the size of the effect. Was this experiment sufficiently powered to reliably detect an effect at -60 mV (which is smaller than the effect at 5 mV)? I would not be sure to "suggest that SCI does not greatly affect Nav1.9", especially with a p-value of 0.08.

We agree with the reviewer that the expression is not precise. We recalculated Nav1.9 current density at 3 min after establishing whole-cell voltage-clamp recording configuration in our previous work and the work from other research groups have demonstrated that Nav1.9 currents run down quickly. Our new data indicates that SCI does not change the fraction of cells producing Nav1.9 currents (Naïve: 36/39 cells; Sham: 42/48 cells; SCI: 45/49 cells; $p = 0.8327$; fisher's exact test). Our new data also indicates that in the cells with Nav1.9 currents, the current density was -0.07 ± 0.11 nA/pF for Naïve, -0.04 ± 0.06 nA/pF for Sham, and -0.07 ± 0.15 nA/pF for SCI, respectively ($p = 0.4833$; one-way ANOVA test). These new data suggest that SCI did not affect Nav1.9. We have added the new data and have corrected the statement in the manuscript. Please see lines 206-210.

- Subgrouping of DRG neurons: the authors chose size as sole distinction criteria. Recent transcriptomic studies suggest that this may not be sufficient to identify those neurons responsible for neuropathic pain. The authors should explain why they used only size and put their choice into context with recent findings from transcriptomic atlases published on mice, rat and other species (e.g. line 73ff, 370ff). Please discuss how to potentially match your choice of DRG subgrouping with that of transcriptomic data.

We certainly agree with the reviewer that cell size is not sufficient to precisely identify those neurons responsible for neuropathic pain. We now mention that patch-seq experiments would help identify which subpopulations of neurons exhibit the increases in Nav1.8 and Nav1.8 resurgent currents. The discussion of these issues has been added in "Discussion" section. Please see lines 332-347.

- o At least subgrouping into tonic vs phasic firing would help. This may have significant impact on the results shown in Fig. 5 and 9 on SCI induced hyperexcitability and its reversal. It may be worth considering additional readouts for hyperexcitability.
- o Line 370ff "Taken together, these findings suggest that the type of spinal cord injury may exert divergent influences on the regulation of VGSC subtype

expression in DRG neurons." Maybe also sensory neuron subtypes play a role, and a more detailed characterization of the neurons you patched is necessary/helpful.

We thank the reviewer for the great suggestion. We tried to subgroup the neurons with tonic or phasic firing. However, it seems difficult to characterize the subgroups of peripheral neurons because phasic and tonic firing patterns can transition into each other in some DRG neurons as the stimulation increases. Some phasic neurons become tonic with increased stimulation and some tonic neurons become phasic with increased stimulations. We did not find a robust way to definitely categorize neurons into one of two groups. So, finally we do not subgroup DRG neurons and simply considered all DRG neurons together.

- The histograms in Figs. 5 and 9 need to be replaced by dot plots.

Thank you. We have replaced the histograms with dot plots.

- Line 345-347/Figure 5/Figure 9: The authors show that SCI does not affect the current threshold but causes hyperexcitability (Figure 5). ZL0177 partially reverses hyperexcitability and increases the current threshold (Figure 9). Line 345-347: "ZL0177 [...]can decrease both TTX-R INaT and INaR, effectively reversing the hyperexcitability of small DRG neurons following SCI." It seems that ZL0177 does not mitigate the SCI-associated neuronal hyperexcitability through the same mechanism as it is caused by. SCI does not affect current threshold, but ZL0177 does. Even though ZL0177 lowers Nav1.8 current density, the lowered excitability might just as well be caused by effects on Nav1.7 or Nav1.9 (the increased current threshold in combination with no change in the RMP points to Nav1.7 in my opinion). How specific is ZL0177? How well is the C-terminus conserved across Nav1.7/Nav1.8/Nav1.9? Could the effects also be mediated by affecting Nav1.7? A change in threshold suggests effects of ZL on currents active during subthreshold depolarization, not so much currents that support repetitive firing as used as readout for hyperexcitability in this study. Please explain this discrepancy and discuss.

We certainly agree that this is important to discuss. Nav1.7, Nav1.8 and Nav1.9 are all critical to regulate DRG neuron excitability. We concentrated on the impact of TTX-R sodium currents because the effects of SCI on TTX-S VGSCs were inconsistent when compared to the naïve and sham groups (Fig. 1F). DRG neuron excitability was measured after adding TTX into the bath solution to block all TTX-S sodium channels including Nav1.7. Therefore, the effects of ZL0177 we determined in this study were not mediated by affecting Nav1.7. ZL0177 was originally reported to disrupt the interaction between Nav1.6 and FHF4 by docking at the channel C-terminus. According to the reference (Liu et al., 2019), the ZL0177 binding site is conserved at the corresponding position among Nav1.6, Nav1.7, Nav1.8 and Nav1.9. It seems highly likely that ZL0177 displays low selectivity to the four sodium channel subtypes. Our new data showed that as observed in Nav1.6 and Nav1.8, 30 μ M ZL0177 also significantly inhibited 75% of Nav1.9 currents in DRG neurons (DMSO: 0.081 ± 0.128 nA/pF, $n = 18$; ZL0177: 0.020 ± 0.043 , $n = 22$; $p = 0.043$). The ZL0177-induced inhibition of SCI DRG neuron hyperexcitability was caused by affecting Nav1.8 as well as Nav1.9. We have added the new data and the discussion in the revised manuscript. Please see lines 269-270, 398-412.

Minor:

- There is a recent paper by Laezza group using FHF4 blockers (<https://pubmed.ncbi.nlm.nih.gov/39747162/>), please discuss and explain the difference in substances used. Also in terms of shifts of activation (line 305ff).

Thank you. We have added the comparison about the compounds 1028 and ZL0177 in the "Discussion" section. Please see lines 392-396.

- Line 273-274: "Average firing frequency is 1.0 {plus minus} 0.2 for the naïve group, 1.0 {plus minus} 0.1 for the sham group, and 7.0 {plus minus} 2.6 for the SCI group (p = 0.0447; one-way ANOVA)." Unit of frequency is missing.

We thank the reviewer for pointing out the oversight. We calculated the number of action potentials firing that were evoked within 2 s. We changed the statement to be "the number of evoked action potentials". Please see line 252.

- Throughout the "results"-section: Sometimes the statistical test is provided after the p-value, sometimes it is not.

We thank the reviewer for pointing out the oversight. We have removed the statistical test in the "Results" section.

- Line 361f: please compare to results from the literature.

Thank you. We have compared our results to those from the literature. Please see lines 301-305.

Referee #2:

Evidence continues to increase for major contributions of hyperactivity in primary nociceptors to persistent pain in various pain conditions, including central neuropathic pain after spinal cord injury (SCI). However, the mechanisms underlying pain-driving nociceptor hyperexcitability are not well understood. The rigorous, well-designed experiments in this manuscript significantly extend our understanding of the contributions of Nav1.8 to the hyperexcitability of probable nociceptors reported to be critical for SCI-induced enhancement of evoked and spontaneous pain-like behavior in rats after SCI (Yang et al. 2014). The authors provide the first independent confirmation that nociceptor hyperexcitability after SCI involves enhanced Nav1.8 function. Even more important, they have revealed that the enhanced Nav1.8 function involves a substantial increase in resurgent current (TTX-R INaR; which does not occur for the TTX-sensitive currents that are probably conducted through Nav1.7 and Nav1.6 channels) and that nociceptor hyperexcitability after SCI can be suppressed by a drug (ZL0177) that they have shown to inhibit FHF4A-mediated TTX-R INaR. Thus, this study adds significantly to the mechanistic understanding of an important mechanism for SCI pain. Given that nociceptor hyperactivity is being found to contribute to many other pain conditions, these findings are likely to have implications beyond SCI. I have a few suggestions to strengthen the manuscript.

We appreciate the reviewer for positive comments and provide thoughtful suggestions to improve the manuscript.

A general suggestion is to discuss more explicitly how the SCI-induced increase in TTX-R INaR may alter DRG neuron function. First, it is worth mentioning why most of the small DRG neurons may be nociceptors, which strengthens the implications of the SCI-induced hyperexcitability for driving pain (e.g., nociceptors are enriched in small DRG neurons and in Nav1.8-expressing DRG neurons, previous studies using similar dissociation conditions found SCI-induced hyperexcitability primarily in capsaicin-sensitive DRG neurons as well as in IB4+ neurons).

Thank you. We have discussed why most of small DRG neurons may be nociceptors. Please see lines 332-347.

Second, is the enhanced TTX-R I_{NaR} so brief that it would only be expected to promote relatively high-frequency bursting (e.g., ~5 Hz, as in Fig.9A), or could it contribute to lower-frequency repetitive firing or ongoing activity? In particular, is the decay time constant for TTX-R I_{NaR} sufficiently long that it could contribute to the low-frequency nociceptor spontaneous activity (SA, often 0.1-2 Hz) that has been associated with SCI. This would need a comment about why the authors did not find SCI-induced SA under their conditions, which differ in multiple ways from those in prior nociceptor SA studies (e.g., female rather than male rats, 2 weeks vs {greater than or equal to} 4 weeks post-SCI test time, 5 mM rather than 3 mM K^+ in the extracellular solution, 175 vs 150 Kdynes spinal impact - with the impact for a 0-s (?) dwell time vs 1-s dwell time).

We expect that the enhanced TTX-R I_{NaR} , primarily generated by Nav1.8, can contribute to both relatively high-frequency firing and lower-frequency repetitive firing in Nav1.8-positive neurons. However, the current may play a limited role in ongoing activity. This might be due to the unique voltage dependence of activation of Nav1.8 I_{NaR} . Nav1.8 I_{NaR} is evoked at -45 mV - +20 mV and peaks at -20 mV, a potential much more positive than the resting membrane potential. Our previous work clearly showed that siRNA-mediated reduction of Nav1.8 I_{NaR} significantly decreases firing frequency of action potentials but fails to change the fraction of DRG neurons with spontaneous activity. In this study, we did not find the change in spontaneous activity for two possible reasons: first, we concentrated on TTX-R sodium channels and measured excitability of DRG neurons pretreated with TTX, which blocked all TTX-S sodium currents including Nav1.7 currents. Nav1.7 is frequently co-expressed with Nav1.8 in DRG neuron. Nav1.7 current can amplify subthreshold depolarizations and bringing them to the threshold. Second, DRG neurons were extracted at two weeks, not four weeks, after SCI surgery. It is likely that four weeks would allow dysregulation of more ion channels or other proteins to contribute to neuronal spontaneous activity than two weeks.

The discussion has been added into the revised manuscript. Please see lines 365-371.

Third, some TTX-R I_{NaR} is seen at -50 and -45 mV in Fig. 3D, which is in the range where pharmacological evidence for a partial contribution of Nav1.8 to depolarizing spontaneous fluctuations (DSFs) of membrane potential was reported for mouse DRG neurons (PMID 37862056). It may be plausible that TTX-R I_{NaR} also contributes to DSF generation following each AP or burst of APs.

We appreciate the reviewer's prospective suggestion. Many studies support that I_{NaR} is crucial for neurons to maintain high-frequency firing. However, unlike classic sodium current elicited by step depolarizations, I_{NaR} is atypical sodium current that reactivates during a mild repolarization following a brief strong depolarization. Although Nav1.8 TTX-R I_{NaR} is reactivated during repolarizing potentials ranging from -45 mV to +20 mV, it seems that Nav1.8 does not produce I_{NaR} at resting membrane potential or before the channel is activated by strong depolarization. Due to this special condition for activation of Nav1.8 I_{NaR} , we assume that TTX-R I_{NaR} (Nav1.8) may not contribute to DSF generation at resting membrane potentials. In our previous work, shRNA-mediated reduction of Nav1.8 did not alter the percentage of DRG neurons with spontaneous firing. In this study SCI does not significantly alter the fraction of DRG neurons with spontaneous firing either (again with the caveat that TTX is present in these recordings). Therefore, we have not characterized the change in DSFs of membrane potential among naïve, sham and SCI groups.

MINOR COMMENTS

Line 63 - It is not self-evident why pain below and at the SCI level suggests abnormal peripheral inputs. Please clarify.

We agree that the statement is not accurate. We have removed the sentence “Firstly, the neuropathic pain observed in SCI patients typically occurs at or below the level of the injury”.

Line 66 - The Siddall 1995 reference is inappropriate - they did not do any experiments on SCI-induced SA in lumbar DRG neurons, none of which were reported until 2010.

Thank you. We have been removed the Siddall 1995 reference.

Line 125 - Which collagenase and protease were used?

The enzymes used are collagenase type I (CAT#: LS004194) and neutral protease (CAT#: LS02104). They are both purchased from Worthington. We have added the information into the Method section. Please see lines 140-141.

Line 137 - Do ND7/23 express endogenous Nav1.9 current?

ND7/23 cells do not express endogenous TTX-R (Nav1.8 or Nav1.9) currents but TTX-sensitive sodium currents. Please see line 153-154.

Line 173 - The unit of statistical analysis is the individual cell or neuron. How many neurons were recorded from each rat?

6-16 cells were recorded from each rat. Please see line 193.

Fig. 3C, Fig.8A - It is impossible to have 150% of the neurons express any kind of current. End the Y-axis at 100%.

Thank the reviewer for pointing out the oversight. We have ended the Y-axis at 100% in Fig. 3C and Fig. 8A.

Fig.6 - It is surprising to have Nav1.8 currents generated by a 10 or 30 mV depolarization from a holding potential of -100 mV. Are these the absolute voltages rather than changes in voltage? If not, a comment about why this occurs in these cells would be helpful.

Sorry for the confusion. The voltage dependence of activation of Nav1.8 I_{NaT} was more positive in ND7/23 cells than in DRG neurons. A depolarization to +10-mV was employed because the current peaked at approximately +10 mV in ND7/23 cells. The depolarization to +30-mV, followed by a repolarizing potential, is a standard protocol for inducing I_{NaR} . The +30-mV depolarization was employed to activate sodium channels so that I_{NaR} mediators could occupy channel pore. To clarify, we have added the protocol as inset in Fig. 6.

Line 396 - Add "Purkinje" before "neurons" for clarity.

Thank you. “Purkinje” has been added before “neurons”.

Line 436 - Correct to Yang et al., 2014.

Thank you. The reference has been corrected.

Dear Dr Cummins,

Re: JP-RP-2025-288215R1 "Contusive spinal cord injury causes Nav1.8 dysfunction to upregulate small sensory neuron excitability" by Yucheng Xiao, Yanling Pan, Naikiu Liu, and Theodore R Cummins

Thank you for submitting your manuscript to The Journal of Physiology. It has been assessed by a Reviewing Editor and by 2 expert referees and we are pleased to tell you that it is acceptable for publication following satisfactory revision.

REVISION CHECKLIST:

We look forward to receiving your revised submission.

Yours sincerely,

Nathan Schoppa
Senior Editor
The Journal of Physiology

REQUIRED ITEMS

Papers must comply with the Statistics Policy: https://jp.msubmit.net/cgi-bin/main.plex?form_type=display_requirements#statistics.

In summary:

- If $n \leq 30$, all data points must be plotted in the figure in a way that reveals their range and distribution. A bar graph with data points overlaid, a box and whisker plot or a violin plot (preferably with data points included) are acceptable formats.
- If $n > 30$, then the entire raw dataset must be made available either as supporting information, or hosted on a not-for-profit repository, e.g. FigShare, with access details provided in the manuscript.
- 'n' clearly defined (e.g. x cells from y slices in z animals) in the Methods. Authors should be mindful of pseudoreplication.
- All relevant 'n' values must be clearly stated in the main text, figures and tables.
- The most appropriate summary statistic (e.g. mean or median and standard deviation) must be used. Standard Error of the Mean (SEM) alone is not permitted.
- Exact p values must be stated. Authors must not use 'greater than' or 'less than'. Exact p values must be stated to three significant figures even when 'no statistical significance' is claimed.

EDITOR COMMENTS

Reviewing Editor:

The revised manuscript has been assessed by the 2 reviewers who overall consider that the manuscript has been improved. However, there are some points raised by both the referees that need to be addressed - including reanalysis/consideration of the current clamp dataset (referee 2)

As previously requested, p values should be added to figure legends.

Senior Editor:

Thank you for submitting your thoroughly revised manuscript to Journal of Physiology. The two original referees and reviewing editor found that the work is now substantially improved. The two referees (especially Referee 2) raised some points that will need to be addressed before final acceptance of the study, including reanalysis/consideration of the current clamp dataset. I have couple of other points to make about statistics:

1. As Senior Editor, I had previously requested that p-values be reported in the figure legends. I see that you have added them onto the figure panels (when there was significance) and this is adequate. So, I do not think that they need to be

added to the legend at this point.

2. There is a question about how you addressed multiple comparisons, most relevant to Figure 9B but I think elsewhere too. Please be clear in the Statistics section of the Methods that you corrected the p-values for multiple comparisons.

REFEREE COMMENTS

Referee #1:

The manuscript has improved a lot, and the authors addressed almost all comments. Too bad that subgrouping by firing patterns was not successful. The line numbers mentioned in the rebuttal letter are off by some numbers, thus, it is not straightforward to identify the spots in the manuscript.

One minor thing:

Discussion line 330: please mention that the "small" fraction, is actually 30% (not so small, after all).

Referee #2:

The authors have met my previous concerns about this study that adds important information to our knowledge of SCI-induced persistent pain mechanisms. However, I have several additional suggestions and questions.

Graphical abstract - the illustrative recordings are not typical for dissociated small DRG neurons. The AHPs are too small. Compare to the illustrations in Figs. 5 and 9 that show actual DRG neuron recordings.

Line 89. "On the other hand" makes no sense. Replace with "In addition" or an equivalent.

Line 124. The data don't distinguish onset from early maintenance. Delete "the onset of".

Line 140. Most users of the Infinite Horizon impactor set a dwell time. If no dwell time was set, this should be reported (e.g., "0-second dwell time").

Line 201. The details of rheobase measurement should be given (current pulse duration, pA increments, time between steps).

Line 319. The conclusion that SCI had no effect on RMP is not convincing because the authors have not set a cut-off for acceptable RMP values (many labs use a -40 mV cutoff). The data may include values from cells that are excessively leaky. What do the data look like when a stated criterion for acceptable neurons is used (such as {less than or equal to} -40 mV RMP)?

Line 320. A threshold current, not "potential", has been measured, and with the long pulses used, this corresponds to rheobase.

Line 385. It appears that 0.5% DMSO (a high concentration for electrophysiology) has depolarized RMP in the experiments summarized in Fig. 9. The medians and ranges of RMP are clearly less than shown in Fig. 5. While there is an effect of ZL0177 vs DMSO, this complication should be mentioned.

Line 400. Correct to Yang et al., 2014.

Line 423. The Coward reference is not optimal, because the DRG neurons examined were axotomized by avulsion. Few if any L4 C-fiber DRG neurons can be axotomized by a vertebral T10 contusion.

Line 447. TRPV1 is more relevant for nociceptors than TRPM8, especially since capsaicin sensitivity is mentioned in the next sentence.

Line 475. The induction of SA in lumbar DRG neurons is not just "a previous finding". It has been replicated in multiple papers using rats and mice.

Line 478. The presence of TTX during current clamp recordings is the most obvious explanation for lack of SCI-induced SA in this study (the presence of TTX apparently was not mentioned in the previous manuscript). However, it also adds to the question about relatively depolarized RMPs found in the control conditions. By blocking Nav1.7, TTX would be expected to produce some hyperpolarization, not depolarization, but the Naïve group RMPs are depolarized compared with previous Naïve RMPs in SCI studies using similar conditions.

END OF COMMENTS

EDITOR COMMENTS

Reviewing Editor:

The revised manuscript has been assessed by the 2 reviewers who overall consider that the manuscript has been improved. However, there are some points raised by both the referees that need to be addressed - including reanalysis/consideration of the current clamp dataset (referee 2)

Response: Thank you. We have responded below to all the points raised by the editors and reviewers. In this response to the reviewers we reference line numbers from the highlighted word document that we uploaded. When the J Physiol website generates a pdf version, the length of the document grew by 5 lines overall. I am not sure if that always happens, but we apologize for any confusion this creates if a reviewer is looking at the pdf version and not the word version.

As previously requested, p values should be added to figure legends.

Response: Per the comment below from the Senior Editor, we confirm that p values are reported in the document, either in the figure legends, figure panels or text. We tried to place these appropriately and not redundantly.

Senior Editor:

Thank you for submitting your thoroughly revised manuscript to Journal of Physiology. The two original referees and reviewing editor found that the work is now substantially improved. The two referees (especially Referee 2) raised some points that will need to be addressed before final acceptance of the study, including reanalysis/consideration of the current clamp dataset. I have couple of other points to make about statistics:

1. As Senior Editor, I had previously requested that p-values be reported in the figure legends. I see that you have added them onto the figure panels (when there was significance) and this is adequate. So, I do not think that they need to be added to the legend at this point.

Response: Thanks. We confirm that p values are reported in the figure legends or panels.

2. There is a question about how you addressed multiple comparisons, most relevant to Figure 9B but I think elsewhere too. Please be clear in the Statistics section of the Methods that you corrected the p-values for multiple comparisons.

Response: Thank you. In Figs. 5B and 9B, multiple comparisons were carried out among different groups at the same injected current. The statement has been added in the Statistics section of the methods. Please see lines 196-197.

REFEREE COMMENTS

Referee #1:

The manuscript has improved a lot, and the authors addressed almost all comments. Too bad that subgrouping by firing patterns was not successful. The line numbers mentioned in the rebuttal letter are off by some numbers, thus, it is not straightforward to identify the spots in the manuscript.

Response: Thank you. Our apologies for the line number issue – we submitted a marked version with tracked changes and the unmarked version and because of deletions the line numbers in these two versions did not match up. For this revision we highlight changes in red but do not show deletions, so the line numbers in the two versions should be the same. In addition, we note that the uploaded word files and the generated pdf files end up with different line numbering. We based the line numbers in this response to the reviewers on the line number in the word document. When the submission process generates the pdf versions, the length of the document grew by 5 lines overall, so numbers in the pdf versions may be a bit off.

One minor thing:

Discussion line 330: please mention that the "small" fraction, is actually 30% (not so small, after all).

Response: We agree with the reviewer. “30%” has been added.
Please see line 331 in the red-marked document.

Referee #2:

The authors have met my previous concerns about this study that adds important information to our knowledge of SCI-induced persistent pain mechanisms. However, I have several additional suggestions and questions.

Graphical abstract - the illustrative recordings are not typical for dissociated small DRG neurons. The AHPs are too small. Compare to the illustrations in Figs. 5 and 9 that show actual DRG neuron recordings.

Response: We agree with the reviewer. The illustrative curves of action potentials have been replaced with those recorded in the experiments.
Please see the new abstract figure.

Line 89. "On the other hand" makes no sense. Replace with "In addition" or an equivalent.

Response: Thanks. “On the other hand” has been replaced with “In addition”.
Please see line 78.

Line 124. The data don't distinguish onset from early maintenance. Delete "the onset of".

Response: Thanks. “the onset of” is deleted.
Please see line 112.

Line 140. Most users of the Infinite Horizon impactor set a dwell time. If no dwell time was set, this should be reported (e.g., "0-second dwell time").

Response: Thanks. We checked our record and found that no dwell time was set. Therefore, “0-s dwell time” has been added in the method.
Please see line 128

Line 201. The details of rheobase measurement should be given (current pulse duration, pA increments, time between steps).

Response: Thanks. The current threshold (rheobase) was measured by a 1-ms injection of step current,

which ranged from 0 to 3000 pA in 100-pA incremental steps with an interval of 5 s. The details have been added in the legends of Fig. 5D and Fig. 9D.

Please see lines 616-617 and 645.

Line 319. The conclusion that SCI had no effect on RMP is not convincing because the authors have not set a cut-off for acceptable RMP values (many labs use a -40 mV cutoff). The data may include values from cells that are excessively leaky. What do the data look like when a stated criterion for acceptable neurons is used (such as {less than or equal to} -40 mV RMP)?

Response: We appreciate the reviewer's expertise. As suggested, we used a -40-mV cutoff in Fig. 5D and the neurons with RMP > -40 mV were excluded. The RMPs were -55.1 ± 12.5 mV for the Naïve group, -51.5 ± 10.4 mV for the Sham group and -49.9 ± 5.9 mV for the SCI group, respectively ($p = 0.1809$). This does not change the conclusion that SCI did not significantly affect RMP of DRG neurons.

Response: Please see lines 612-615.

Line 320. A threshold current, not "potential", has been measured, and with the long pulses used, this corresponds to rheobase.

Response: Thank you. "potential" has been replaced with "current".

Please see lines 258 and 645.

Line 385. It appears that 0.5% DMSO (a high concentration for electrophysiology) has depolarized RMP in the experiments summarized in Fig. 9. The medians and ranges of RMP are clearly less than shown in Fig. 5. While there is an effect of ZL0177 vs DMSO, this complication should be mentioned.

Response: Thank you. The 0.5% concentration of DMSO was used because in our experiments ZL0177 could be completely dissolved in 0.5% DMSO, but not in DMSO below this concentration. This concentration of DMSO does seem to increase leak current under the whole cell recording configuration a bit so that the RMPs in Fig. 9D are less than those in Fig. 5D. In Fig. 9D, the minimum acceptable RMP was set to -35 mV. The complication has been mentioned in the Results section.

Please see lines 288-290 and 643-644.

Line 400. Correct to Yang et al., 2014.

Response: Thanks. The reference has been corrected.

Please see line 299.

Line 423. The Coward reference is not optimal, because the DRG neurons examined were axotomized by avulsion. Few if any L4 C-fiber DRG neurons can be axotomized by a vertebral T10 contusion.

Thank you. This is reasonable. The reference and the related statement have been removed.

Line 447. TRPV1 is more relevant for nociceptors than TRPM8, especially since capsaicin sensitivity is mentioned in the next sentence.

Response: Thank you. "TRPM8" has been deleted.

Please see line 341.

Line 475. The induction of SA in lumbar DRG neurons is not just "a previous finding". It has been replicated in multiple papers using rats and mice.

Response: We agree with the reviewer that “a previous finding” was not adequate. We have replaced it with “previous findings”. Two more references (Wu et al., 2013; Yang et al. 2014) have been added. Please see lines 368-369.

Line 478. The presence of TTX during current clamp recordings is the most obvious explanation for lack of SCI-induced SA in this study (the presence of TTX apparently was not mentioned in the previous manuscript). However, it also adds to the question about relatively depolarized RMPs found in the control conditions. By blocking Nav1.7, TTX would be expected to produce some hyperpolarization, not depolarization, but the Naïve group RMPs are depolarized compared with previous Naïve RMPs in SCI studies using similar conditions.

Response: Thank you. The relatively depolarized RMPs might be caused partially by not setting a cut-off of -40 mV. In the previous manuscript, we calculated the RMPs from all small DRG neurons we patched. After setting a -40-mV cut-off, the RMPs ranged from -85.7 mV to -40.0 mV for the Naïve group, and the average RMP was -55.1 ± 12.5 mV, which are close to the values reported in small isolated DRG neurons in many previous studies (e.g. Bedi et al., 2010; Zhang et al., 2004, Brain Research; Renganathan et al., 2002, J Neurophysiol; Huang et al., 2017, J Clin Invest). We assume that blocking Nav1.7 would have a limited influence in regulation of resting membrane potentials because Nav1.7 begins to activate at approximately -45 mV, which is 10-mV more negative than RMPs. Therefore, few Nav1.7 channels would be expected to activate at resting membrane potentials. Nav1.7 can produce larger ramp currents than other sodium channel subtypes. Indeed, while previous studies support that Nav1.7 plays important roles in helping setting threshold for action potential firing, dynamic-clamp experiments from the Waxman lab have indicated that WT Nav1.7 channels have minimal impact on resting membrane potential. We have a brief discussion of the role of Nav1.7 – Nav1.9 in action potential firing in the Discussion section, but in an effort to keep the discussion at a reasonable length we do not go into an extensive discussion of previous studies that examined the impact of Nav1.7 currents on RMP.

Please see lines 311-316.

Dear Dr Cummins,

Re: JP-RP-2025-288215R2 "Contusive spinal cord injury causes Nav1.8 dysfunction to upregulate small sensory neuron excitability" by Yucheng Xiao, Yanling Pan, Naikiu Liu, and Theodore R Cummins

Thank you for submitting your manuscript to The Journal of Physiology. It has been assessed by a Reviewing Editor and by 1 expert referee and we are pleased to tell you that it is acceptable for publication following satisfactory revision.

REVISION CHECKLIST:

We look forward to receiving your revised submission.

Yours sincerely,

Nathan Schoppa
Senior Editor
The Journal of Physiology

REQUIRED ITEMS

- You must start the Methods section with a paragraph headed Ethical approval (https://jp.msubmit.net/cgi-bin/main.plex?form_type=display_requirements#methods).

Research must comply with The Journal's policies regarding animal experiments (<https://physoc.onlinelibrary.wiley.com/hub/animal-experiments>) and adherence to these policies must be stated in the manuscript.

Authors should confirm in their Methods section that their experiments were carried out according to the guidelines laid down by their institution's animal welfare committee, including an ethics approval reference number. The Methods section must contain a statement about access to food, water and housing, details of the anaesthetic regime: anaesthetic used, dose and route of administration, and method of killing the experimental animals.

EDITOR COMMENTS

Reviewing Editor:

The authors have thoroughly addressed reviewer comments. There is one outstanding minor change to text requested.

Please also see 'Required Items' above.

Senior Editor:

Thank you for submitting your revised manuscript. It has been reviewed by one of the original referees and reviewing editor, and the prior concerns have been well-addressed and the overall assessment is very positive. A few minor additional points however have been identified that will require one more revision. They are listed below. The first two points in principle could have been handled in the proofs stage, but the third, I think, requires another quick round of revision and review. The senior editor will review the revised manuscript.

(1) The referee raised a point about the use of the word "rheobase" in the legend of Figure 9 that needs to be addressed.

(2) The senior editor comments about the prior submission included a question about whether, in the statistical analysis

related to Fig. 9B (also applies to Fig. 5B), the authors performed a correction for multiple comparisons. The revision includes some clarification about the analysis but does not appear to be explicit about whether a correction was performed (e.g., in the p-values themselves or the p-value that is considered to be significant).

(3) There appears to be information missing about analgesics that were used as part of the post-operative care of the rats.

REFEREE COMMENTS

Referee #2:

The authors have met my additional concerns and answered all my questions well.

I have just one very minor correction. In the legend for Fig. 9 (line 649) they write "threshold current (rheobase)." Remove the word "rheobase" because a 1-ms pulse is much too brief for assessing rheobase (which is defined as the lowest threshold current for any duration of depolarization).

END OF COMMENTS

Reviewing

Editor:

The authors have thoroughly addressed reviewer comments. There is one outstanding minor change to text requested.

Please also see 'Required Items' above.

Senior Editor:

Thank you for submitting your revised manuscript. It has been reviewed by one of the original referees and reviewing editor; and the prior concerns have been well-addressed and the overall assessment is very positive. A few minor additional points however have been identified that will require one more revision. They are listed below. The first two points in principle could have been handled in the proofs stage, but the third, I think, requires another quick round of revision and review. The senior editor will review the revised manuscript.

(1) The referee raised a point about the use of the word "rheobase" in the legend of Figure 9 that needs to be addressed.

Response: Thank you. As suggested by the reviewer#2, we have deleted “rheobase” in the legend of Fig. 9. Please see line 654.

(2) The senior editor comments about the prior submission included a question about whether, in the statistical analysis related to Fig. 9B (also applies to Fig. 5B), the authors performed a correction for multiple comparisons. The revision includes some clarification about the analysis but does not appear to be explicit about whether a correction was performed (e.g., in the p-values themselves or the p-value that is considered to be significant).

Response: We appreciate the senior editor’s expertise. We now have shown the statistical methods and the post-hoc tests for multiple comparisons for figures 1, 3, 4, 5, 7, 8 and 9, in which statistical analysis was performed. Please see the legends of these figures. All p values have been shown in the figures. The statement that $p < 0.05$ indicated a significant change has also been added. In Fig. 5B (Fig. 3 and Fig. 4 as well), the corrected p-values after multiple comparison have been shown at the corresponding positions in the “Results” section. In Fig. 9B, there are only two groups (DMSO and ZL0177) within the same injected current. Therefore, the p values are calculated using t-test and no correction for multiple comparisons is performed.

Please see lines 261-262, 579-580, 599, 600, 603-604, 612, 614-615, 620-621, 628-629, 644, 646-647, 649, and 655-656.

(3) There appears to be information missing about analgesics that were used as part of the post-operative care of the rats.

Response: Thanks. The information has been added in the section "*Animals and Contusive SCP*" of Materials and Methods.

Please see lines 133-138.

REFeree COMMENTS

Referee #2:

The authors have met my additional concerns and answered all my questions well. I have just one very minor correction. In the legend for Fig. 9 (line 649) they write "threshold current (rheobase)." Remove the word "rheobase" because a 1-ms pulse is much too brief for assessing rheobase (which is defined as the lowest threshold current for any duration of depolarization).

Response: Thanks. "rheobase" has been removed in the legend of Fig. 9. Please see line 654.

REQUIRED ITEMS

- You must start the Methods section with a paragraph headed Ethical approval (https://jp.msubmit.net/cgi-bin/main.plex?form_type=display_requirements#methods).

Research must comply with The Journal's policies regarding animal experiments (<https://physoc.onlinelibrary.wiley.com/hub/animal-experiments>) and adherence to these policies must be stated in the manuscript.

Authors should confirm in their Methods section that their experiments were carried out according to the guidelines laid down by their institution's animal welfare committee, including an ethics approval reference number. The Methods section must contain a statement about access to food, water and housing, details of the anaesthetic regime: anaesthetic used, dose and route of administration, and method of killing the experimental animals.

Response: In the previous version the Methods section was indeed started with a paragraph headed Ethical approval. We revised the language slightly in this revision to conform with other recent examples. The ethics approval reference number is included as required. We have checked that the methods section contains the required statements and we have added additional information regarding the analgesics as described above.

Dear Professor Cummins,

Re: JP-RP-2025-288215R3 "Contusive spinal cord injury causes Nav1.8 dysfunction to upregulate small sensory neuron excitability" by Yucheng Xiao, Yanling Pan, Naikiu Liu, and Theodore R Cummins

We are pleased to tell you that your paper has been accepted for publication in The Journal of Physiology.

Yours sincerely,

Nathan Schoppa
Senior Editor
The Journal of Physiology

If you would like to receive our 'Research Roundup', a monthly newsletter highlighting the cutting-edge research published in The Physiological Society's family of journals (The Journal of Physiology, Experimental Physiology, Physiological Reports, The Journal of Nutritional Physiology and The Journal of Precision Medicine: Health and Disease), please click this link, fill in your name and email address and select 'Research Roundup':

<https://www.physoc.org/journals-and-media/membernews>

- **TRANSPARENT PEER REVIEW POLICY:** To improve the transparency of its peer review process, The Journal of Physiology publishes online as supporting information the peer review history of all articles accepted for publication. Readers will have access to decision letters, including Editors' comments and referee reports, for each version of the manuscript as well as any author responses to peer review comments. Referees can decide whether or not they wish to be named on the peer review history document.
- You can help your research get the attention it deserves! Check out Wiley's free Promotion Guide for best-practice recommendations for promoting your work at: www.wileyauthors.com/eeo/guide. You can learn more about Wiley Editing Services which offers professional video, design, and writing services to create shareable video abstracts, infographics, conference posters, lay summaries, and research news stories for your research at: www.wileyauthors.com/eeo/promotion.
- **IMPORTANT NOTICE ABOUT OPEN ACCESS:** To assist authors whose funding agencies mandate public access to published research findings sooner than 12 months after publication, The Journal of Physiology allows authors to pay an Open Access (OA) fee to have their papers made freely available immediately on publication.

EDITOR COMMENTS

The authors have adequately addressed all remaining concerns. We appreciate the extra attention to the final details. Congratulations! The paper is now acceptable for publication.